# Forecasting monthly runoff in a glacierized catchment: A comparison of extreme gradient boosting (XGBoost) and deep learning models

**Mohammed Majeed Hameed**[1]*, **Adil Masood**[2], **Aadil hamid**[2], **Ahmed Elbeltagi**[3]*, **Siti Fatin Mohd Razali**[4], **Ali Salem**[5,6]*

**1** Upper Euphrates Center for Sustainable Development Research, University of Anbar, Ramadi, Iraq, **2** Department of Natural and Applied Sciences, TERI School of Advanced Studies, New Delhi, India, **3** Agricultural Engineering Department, Faculty of Agriculture, Mansoura University, Mansoura, Egypt, **4** Department of Civil Engineering, Faculty of Engineering and Built Environment, Universiti Kebangsaan Malaysia (UKM), Bangi, Malaysia, **5** Civil Engineering Department, Faculty of Engineering, Minia University, Minia, Egypt, **6** Structural Diagnostics and Analysis Research Group, Faculty of Engineering and Information Technology, University of Pécs, Pécs, Hungary

* mohmmag1@gmail.com (MMH); ahmedelbeltagy81@mans.edu.eg (AE); salem.ali@mik.pte.hu (AS)

## Abstract

Accurate monthly runoff forecasting is vital for water management, flood control, hydropower, and irrigation. In glacierized catchments affected by climate change, runoff is influenced by complex hydrological processes, making precise forecasting even more challenging. To address this, the study focuses on the Lotschental catchment in Switzerland, conducting a comprehensive comparison between deep learning and ensemble-based models. Given the significant autocorrelation in runoff time series data, which may hinder the evaluation of prediction models, a novel statistical method is employed to assess the effectiveness of forecasting models in detecting turning points in the runoff data. The performance of Extreme Gradient Boosting (XGBoost) was compared with long short-term memory (LSTM) and random forest (RF) models for one-month-ahead runoff forecasting. The study used 20 years of runoff data (2002–2021), with 70% (2002–2015) dedicated for training and calibration, and the remaining data (2016–2021) for testing. The findings for the testing phase results show that the XGBoost model achieves the best accuracy, with $R^2$ of 0.904, $RMSE$ of 1.554 m³/sec, an $NSE$ of 0.797, and Willmott index ($d$) of 0.972, outperforming both the LSTM and RF models. The study also found that the XGBoost model estimated turning points more accurately, obtaining forecasting improvements of up to 22% to 34% compared to LSTM and RF models. Overall, the study's findings are essential for global water resource management, providing insights that can inform sustainable practices to support societies impacted by climate change.

**Data availability statement:** All relevant data are within the paper and its Supporting Information files.

**Funding:** The author(s) received no specific funding for this work.

**Competing interests:** The authors have declared that no competing interests exist.

**Abbreviations:** AAETP, Average absolute error of turning points; ANN, Artificial Neural Network; AR, Autoregressive; ARIMA, Autoregressive Integrated Moving Average; ARMA, Autoregressive Moving Average; BiLSTM, Bidirectional Long Short-Term Memory; CEEMDAN, Complete Ensemble Empirical Mode Decomposition with Adaptive Noise; EANN, Emotional Artificial Neural Network; ELM, Extreme Learning Machine; EOF, Edge-Of-Field; FOEN, Swiss Federal Office for the environment; GBM-Gradient Boosting Machines; GMM, Gaussian Mixture Model; HBV, Hydrologiska Byråns Vattenbalansavdelning; LSSVM, Least Squares Support Vector Machine; LSTM, Long Short-Term Memory; MAE, Mean Absolute Error; NSE, Nash-Sutcliffe Efficiency Index; PACF, Partial Autocorrelation Function; $R^2$, Coefficient of Determination; RF, Random Forest; RMSE, Root Mean Square Error; RNN, Recurrent Neural Network; RNN, Recurrent Neural Network; SHAP, SHapley Additive exPlanations; SSA, Singular Spectrum Analysis; SVM, Support Vector Machine; SVR; U95, Uncertainty at a 95% Confidence Interval; VMD, Variational Mode Decomposition; XGBoost, Extreme Gradient Boostig.

# 1. Introduction

Hydrological Runoff Forecasting plays a crucial role in today's water resource management, regional water resource system, controlling flood water for agriculture and hydrological power projects and environmental impact assessment [1,2]. Runoff denotes the fraction of precipitation or snowmelt that flows over the land surface and eventually confluences with streams, rivers, and other types of fluvial systems. River runoff constitutes a substantial replenishment source for the groundwater basin within the region. Accurate runoff forecasting is a vital engineering endeavor for mitigating the impacts of flood disasters. Improving the precision of runoff predictions remains a formidable challenge in the realm of water resources management research [3]. Various models have been created to predict runoff in Glacierized catchments. For example, the HBV (Hydrologiska Byråns Vattenbalansavdelning) model was utilized for studying the effect of climate change on the runoff and glacier changes in Yushugou Basin, China [4] and to calculate the snowmelt and its share to the overall streamflow in the Panchachuli glacier region of Central Himalaya [5].

Accurate runoff prediction is vital for managing water resources. It also allows for better flood and drought control, reservoir management, and hydropower generation, ultimately leading to more efficient water use [6]. In particular, monthly-scale runoff forecasting is essential for effectively monitoring natural hydrological disasters such as droughts and floods, as these forecasts enable the establishment of early warning systems [7–9]. Also, this allows necessary measures to be taken in advance, mitigating their socio-economic impacts as well as improving water resources management. In general, the models used for runoff prediction are mainly divided into two sections: Process driven, and Data driven models [10]. These models are also widely used for runoff estimation and prediction ([11–13]). Several studies have reported that process-driven models are primarily constructed based on an understanding of runoff development processes [14–16]. These models aim to derive physical parameters that can be used for simulation or prediction tasks. On the other hand, data-driven models solely focus on the input-output relationship without making explicit causal inferences [17]. Also, the process of driven models relies on linear and partial differential equations [18,19]. Runoff prediction methods are primarily categorized into process-driven and data-driven models. Process-driven models like Soil and Water Assessment Tool (SWAT) [20] and Hydrological Simulation Program-Fortran (HSPF) [21] simulate hydrological processes using physical principles. They provide interpretable results but often being computationally intensive and require lots of hydrological and climate variables. While some researchers point out that runoff prediction is a multifaceted hurdle [22], it is influenced by numerous uncertain factors, which include high complexity, non-linear factors, non-stationarity, and dynamism, hence making it difficult to accurately predict runoff by process-driven methods. Moreover, conceptual models require extensive hydrological data, which may not be available in certain regions. Additionally, the implementation of such models is often more expensive and time-consuming. Therefore, various researchers inferred the input-output relationship of data-driven models to be linear and hence predicted runoff by methods like an autoregressive moving average (ARMA) model [23], autoregressive integrated moving average (ARIMA) model [24] and autoregressive (AR) model [25]. Also, other researchers inferred the prediction result could be made better by taking into consideration the non-linear nature [26–28].

Moreover, a variety of other classical machine learning models such as artificial neural networks (ANN) [29], SVM [30], and Random Forest (RF) [31] have been applied for studying runoff prediction. Advanced models based on deep learning have also been developed for various activities involving prediction applications, which include Runoff-prediction [32], prediction of precipitation [33], properties and composition of soil [34], water table [35], and streamflow forecasting [36]. In recent years, machine learning and deep learning models

have garnered significant interest from researchers in constructing highly computationally intensive and accurate forecasting systems. Some researchers compared three boosting models (Extreme Gradient Boosting (XGBoost), Light Gradient-Boosting Machine (Light GBM) & Categorical Boosting (CatBoost)) to forecast daily streamflow in mountainous catchment [37]. The study concluded that the gradient boosting algorithms used in models such as XGBoost, are simple to implement, fast, and robust.

It is important to note that both long short-term memory (LSTM), and XGBoost are commonly used to model several hydrological parameters (e.g., runoff) due to their efficiency in handling time series data [38–40]. The LSTM algorithm is highly effective for modeling time series data, as it captures both short- and long-term dependencies within sequential data. Furthermore, RF is a robust ensemble method that captures complex relationships in runoff data (e.g., reduces) overfitting by averaging across multiple decision trees [41–43]. Besides, the XGBoost model is a powerful gradient-boosting algorithm that excels at modeling non-linear relationships and performs exceptionally well in time series forecasting, particularly with lagged variables due to its advanced optimization and ability to handle structured data.

Some investigations constructed a hybrid model which is formulated by using extreme gradient boosting and Gaussian mixture model, which they named as GMM-XGBoost [29]. They applied this model to predict monthly streamflow at Cuntan & Hankou stations on Yangtze River and the result concluded that the XGBoost-based models performed well and are applicable for streamflow forecasting and are superior to SVM. In a similar work, [44] combined machine learning (ML) models XGBoost and SHAP to develop an explainable ML-based model (KXGBoost) and its results demonstrate a robust data-driven model for runoff prediction. Other scholars investigated the hybrid model created by integration of Fourier transform with LSTM to predict monthly runoff in Brahmani Rive, India [45]. The researchers found that the suggested model outperforms several classical and deep learning models. Some investigations have confirmed that classical ML models can be enhanced by incorporating bio-inspired algorithms and data preprocessing techniques [46–49], thereby improving the models' capability in forecasting runoff. An advanced model such as LSTM with AM (Attention mechanism) was developed for estimating runoff in Hun River, China [50]. It was reported that the model presented high accuracy, indicating its potential in hydrological modeling. Moreover, other researchers [51] demonstrated the framework efficacy of daily edge-of-field (EOF) runoff prediction using the XGBoost algorithm and concluded that the effectiveness of the XGBoost model in runoff prediction is well-supported by various studies emphasizing its accuracy, robustness, and efficiency. Its capability to minimize overfitting while maintaining high predictive accuracy makes it an invaluable tool in hydrological modeling and other environmental prediction tasks. As this field keeps on growing, XGBoost is likely to remain a cornerstone model for researchers and practitioners aiming to enhance predictive accuracy and efficiency in runoff prediction.

Despite the advancement of various models for runoff forecasting conducting in previous research, a significant knowledge gap remains regarding the comprehensive comparative effectiveness of machine learning approaches such as XGBoost, RF, and advanced deep learning models like LSTM. Moreover, the comparison of deep learning models such as LSTM, XGBoost offers an easier interpretation of the significance of predictors. Also, previous research indicates that very few studies have focused on modeling runoff in the Lotschental catchment in Switzerland. A comprehensive study of such catchments is hydrologically significant due to their provision of essential freshwater through glacial melt, particularly in the warmer months. Such catchments are important indicators of climate change, influencing local hydrology and downstream water availability, and they also pose flood risks during melt periods, necessitating effective management strategies. In prior works, researchers interested

in modeling runoff time series data have often overlooked the importance of analyzing how well these predictive models can capture turning points. The turning points are essential for understanding significant shifts in hydrological dynamics.

In our study, we propose using the XGBoost model to forecast monthly rainfall one month ahead in the Lotschental catchment, Switzerland, and compared its performance with other AI model such as LSTM and regression tree model like RF. Runoff generation in mountain regions is affected by cyclogenetic processes [52] and the development of mesoscale circulations [53], which makes the process of runoff prediction extremely challenging. Therefore, developing a reliable projection model for runoff prediction is a difficult task. In order to address this challenge of limited accuracy and reliability in runoff prediction, we developed and compared three state-of-the-art machine learning models to determine the most effective technique for runoff forecasting. Additionally, tuning points in the runoff time series have been detected using a new algorithm. These tuning points are very important in analyzing time series data, as they identify significant shifts in trends (peaks and troughs) within the runoff data, providing valuable insights into the system's underlying dynamics. Thus, the efficiency of the applied forecasting models has been evaluated based on their ability to adequately capture these tuning points. By capitalizing the findings from these studies, this study aims to contribute to the existing knowledge on monthly runoff forecasting in glacierized catchments. Notably, this study is among the first to compare XGBoost, LSTM, and RF models for runoff prediction in glacier catchments, utilizing turning point analysis. The study aims to:

1) Develop models using XGBoost and LSTM for forecasting runoff, focusing on selecting appropriate features using partial autocorrelation function.

2) Compare the performance of XGBoost and LSTM with other ML models based on statistical evaluation metrics such as *RMSE*, *MAE*, $R^2$, and other metrics to assess their predictive accuracy and reliability.

3) Identify and analyze the turning points in monthly runoff time series data and assess the efficiency of forecasting models in accurately predicting these points.

4) Investigate the potential of XGBoost for forecasting runoff by analyzing its predictive capabilities, robustness, and scalability in different hydrological scenarios and datasets.

The comparative analysis between XGBoost and Deep Learning models will provide insights into the strengths and limitations of each approach, ultimately guiding decision-makers in selecting the most effective modeling technique for accurate and reliable runoff predictions in glacier-influenced watersheds.

To achieve the above-mentioned research objectives, four key research questions (RQs) have been developed:

RQ1: How effective are XGBoost and LSTM models in forecasting runoff when using the Partial Autocorrelation Function (PACF) to select relevant features?

RQ2: How does the performance of XGBoost and LSTM models compare to other machine learning models in terms of *RMSE*, *MAE*, and $R^2$ for runoff forecasting?

RQ3: Can XGBoost and LSTM models accurately identify and predict turning points in monthly runoff time series data?

RQ4: How robust and scalable are XGBoost models for runoff forecasting across various hydrological scenarios and datasets?

## 2. Case study location

The Lötschental catchment occupies a unique transitional zone between the Mediterranean climate of the Southern Alps and the maritime climate of the Northern Alps. Located at 46° 24' 59.99" N latitude and 7° 49' 59.99" E longitude (See Fig 1), the catchment sits within the Jungfrau-Aletsch-Bietschhorn region of the Bernese Alps at an elevation of 2624 meters above sea level [54]. The Lonza River, fed by its primary tributaries, the Lang and Anun Glaciers, flows through the valley, with the glaciers themselves forming the eastern and northeastern boundaries of the Lötschental [55]. In recent years, the valley has experienced significant flooding due to increased discharge from mountain rivers in the Bernese Alps, highlighting the region's susceptibility to climate change-induced floods [56]. Previous studies have documented substantial changes in and around the study area due to the climate change [57]. Furthermore, active fault lines underpinning the region give rise to a series of earthquakes.

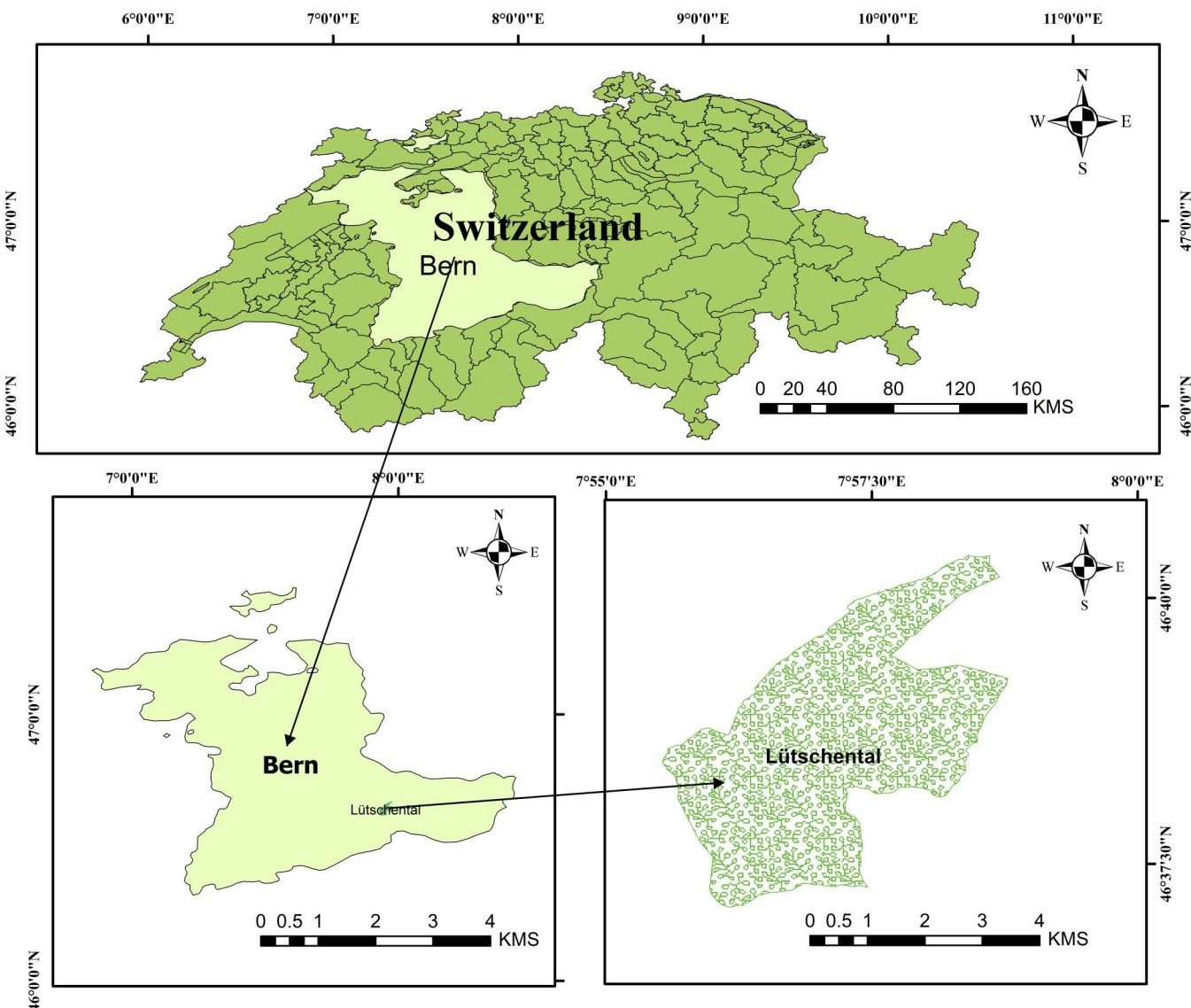

**Fig 1. Case study location (Generated using ArcGIS 10.8.2).**

The faults detected in the Canton of Bern potentially influenced by localized stress conditions, pose a threat to regional stability and enhance the susceptibility of landslides [58]. Also, the 2005 floods drastically affected the region, due to severe destruction in 900 municipalities and complete infrastructure damage in others, as well as some documented displacements within the Canton of Bern [59]

The surging tributaries of the Lonza River have caused substantial erosion and debris flow, underscoring the critical need for accurate runoff forecasting to mitigate flood risks in this area. Daily precipitation and mean air temperature data were acquired from MeteoSwiss for the Blatten, Lötschental station located at 1538 meters above sea level. Corresponding daily runoff (Q) measurements for the Lonza-Blatten hydrometric station were obtained from the Swiss Federal Office for the Environment (FOEN). This station sits at 1520 meters and drains an upstream area of approximately 77 km², with 24.6% glacier coverage. Table 1 summarizes the key statistics of the meteorological and hydrometric data used, while Fig 2 presents a bar chart depicting the monthly distribution of measured runoff during the study period. It is evident that the applied

**Table 1. Statistical description of the monthly runoff records.**

| Month | Maximum m³/sec | Minimum m³/sec | Median m³/sec | Standard deviation m³/sec | Mean m³/sec | Skewness |
|---|---|---|---|---|---|---|
| January | 0.764 | 0.488 | 0.601 | 0.076 | 0.601 | 0.327 |
| February | 0.734 | 0.458 | 0.537 | 0.060 | 0.541 | 1.498 |
| March | 1.013 | 0.444 | 0.634 | 0.160 | 0.673 | 0.627 |
| April | 3.805 | 0.956 | 1.751 | 0.696 | 1.904 | 1.452 |
| May | 8.438 | 3.016 | 5.094 | 1.692 | 5.514 | 0.339 |
| June | 16.887 | 8.201 | 11.606 | 2.555 | 11.919 | 0.627 |
| July | 16.343 | 9.547 | 12.851 | 1.949 | 12.854 | -0.051 |
| August | 14.962 | 6.331 | 11.372 | 1.905 | 11.126 | -0.382 |
| September | 8.565 | 3.103 | 6.590 | 1.234 | 6.317 | -0.691 |
| October | 4.197 | 1.724 | 2.871 | 0.682 | 2.776 | 0.347 |
| November | 2.374 | 0.929 | 1.196 | 0.375 | 1.296 | 1.712 |
| December | 1.007 | 0.615 | 0.774 | 0.103 | 0.780 | 0.425 |

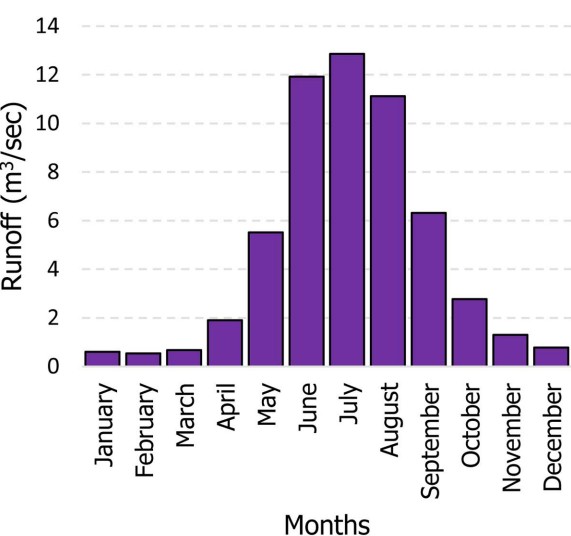

**Fig 2. The Overall measured runoff values for each month.**

case study experienced the highest runoff rates in June, July, and August, with values of 11.919 m³/sec, 12.854 m³/sec, and 11.126 m³/sec respectively. As shown in Fig 2, runoff values demonstrate a clear seasonal pattern, with the highest values recorded in July (likely due to snowmelt) and the lowest values occurring in February. Notably, the data regarding the mean monthly runoff used in this study for Lötschental station are provided in the S Appendix (S1 Table) [60].

## 3. Machine learning models

This study employs three advanced machine learning models: Long Short-Term Memory (LSTM) networks, Extreme Gradient Boosting (XGBoost), and Random Forest (RF) to forecast runoff in the Lotschental catchment in Switzerland. These models are particularly well-suited for runoff forecasting due to their ability to capture complex, nonlinear relationships within hydrological data, handle long-term dependencies, and mitigate overfitting.

 The LSTM algorithm is well-suited for modeling time series data as it effectively captures both short- and long-term dependencies in sequential data. Furthermore, RF is a robust ensemble method that captures complex relationships in runoff data and reduces overfitting by averaging across multiple decision trees. Lastly, XGBoost is a powerful gradient-boosting algorithm that efficiently models non-linear relationships. It performs exceptionally well in time series forecasting, especially with lagged variables, due to its advanced optimization and ability to handle structured data.

### 3.1 LSTM

LSTM is a type of recurrent neural network that processes sequential data using memory cells within the hidden layer [61]. These cells control information flow through forget, input, and output gates (Fig 3). The forget gate discards irrelevant past information, the input gate selects new data to remember, and the output gate determines what processed information affects the next cell. This gating mechanism allows LSTMs to learn long-term dependencies in the sequential data. The LSTM's gated configuration enables long-term information retention over multiple time steps, effectively mitigating the vanishing gradient problem common in classical RNN models [62]. Mathematically, the operation of a memory cell in an LSTM can be described as follows [63]:

$$F_t = \sigma(\omega_f I_t + \bigcup_f h_{t-1} + \beta_F) \tag{1}$$

$$i_t = \sigma(\omega_i I_t + \bigcup_i h_{t-1} + \beta_i) \tag{2}$$

$$O_t = \sigma(\omega_o I_t + \bigcup_o h_{t-1} + \beta_O) \tag{3}$$

$$C_t = F_t {}^\star C_{t-1} + i_t {}^\star \sigma_C\left(\omega_c I_t + \bigcup_C h_{t-1} + \beta_c\right) \tag{4}$$

$$h_t = O_t {}^\star \tanh\left(C_t\right) \tag{5}$$

Where, $F_t$, $i_t$ and $O_t$ represent forget gate, input gate, and output gate, $I_t$ represents the input, $C_t$. is the cell state, $\sigma$, and tanh are the sigmoid and hyperbolic tangent activation functions, $\beta_i$, $\beta_O$, $\beta_F$ represent the biases in the network and $\omega_F$ $\omega_i$, $\omega_O$ and $\omega_c$ are the corresponding weigh for the forget gate, input gate, output gate, and cell state respectively.

 In this study, a deep Long Short-Term Memory (LSTM) neural network architecture was utilized, as depicted in Fig 3b. This architecture comprises an input layer, two LSTM layers (LSTM-1 and LSTM-2), a dense layer, and a lambda layer.

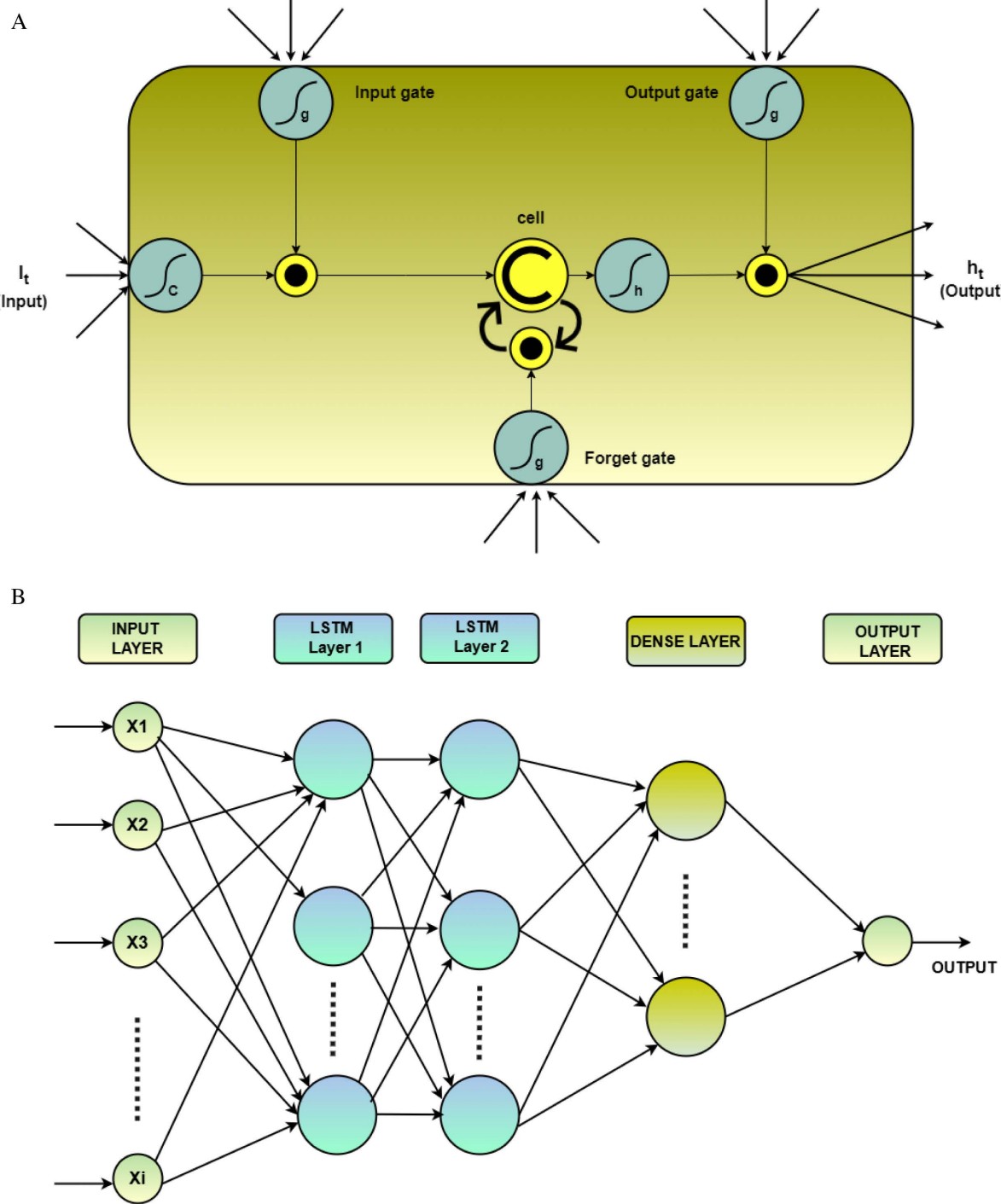

**Fig 3.** a. Typical configuration of LSTM. b. Proposed LSTM model.

The initial LSTM layer (LSTM-1) incorporates numerous units and employs a Rectified Linear Unit (ReLU) activation function to generate a sequence output. Subsequently, the second LSTM layer (LSTM-2) utilizes several units and a ReLU activation function, but without producing a sequence. The dense layer, consisting of n units,

employs a sigmoid activation function. Finally, a lambda layer used to scale the output of the dense layer.

## 3.2 RF

Random forest, a supervised machine learning approach, leverages ensemble learning to improve prediction accuracy. They combine numerous estimator decision trees, each making individual predictions [64] (Fig 4). Training utilizes bagging, where subsets of data are used to grow each tree. This technique, called bootstrap aggregation, reduces overfitting and improves model robustness [65]. In this study, we employ random forest regression, where each tree node considers a random subset of input variables. The final prediction is the average of all individual tree predictions [66]. As the number of trees increases, so does the overall prediction accuracy.

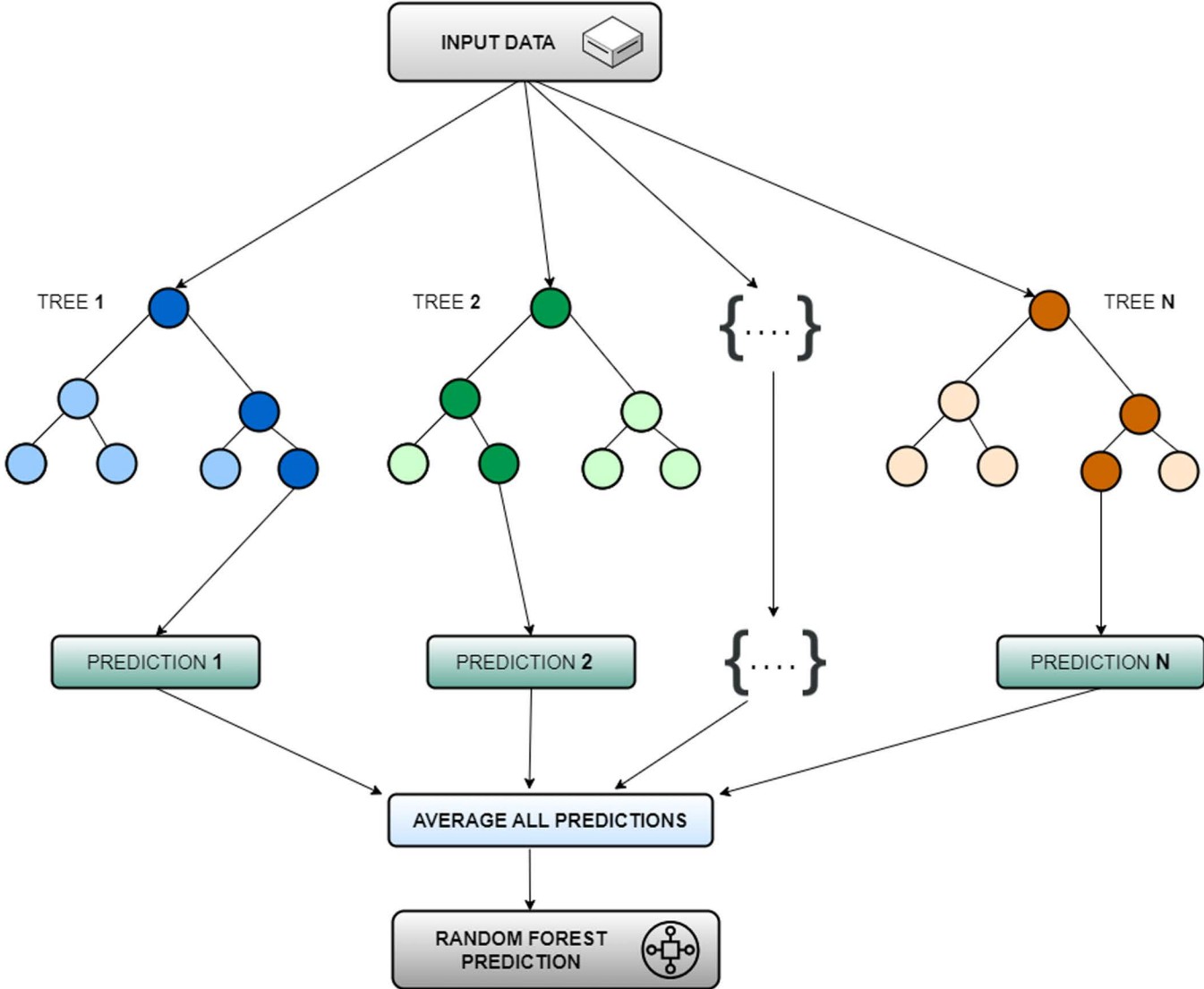

**Fig 4. Random Forest based prediction process.**

### 3.3 XGBoost

The XGBoost model employs the concept of boosting, which aggregates the predictions of multiple weak learners to construct a potent and strong ensemble learner through an additive training approach [67]. This modeling technique, unlike other ML models, is complex with many tunable parameters [68]. Strategically designed to reduce overfitting and prediction variability while enhancing overall prediction accuracy, the technique is widely used in hydrology [69] for various applications, including modeling groundwater salinity, [70] water quality [71], ice phenomena, stream temperatures [37], and runoff prediction [10].

The following outlines the general algorithm for XGBoost applied to regression tasks:

i. Set $\check{O}_i = 0$, $\rightarrow R_i = \check{O}_i - \check{O}$  $\forall\, i$ in the training set (X, y).

ii. For T = 1: N

 1. Development of cost function

$$Obj^{(T)} = \sum_{i=1}^{n}[l(O_i, \check{O}_i^{T-1} + F_T(x_i)] + \Omega(F_T) \tag{6}$$

 2. Train a decision tree ($F_T$) to minimize the objective function ($Obj^{(T)}$) on the training data (X, R)

 3. Update $O$ by incorporating a new tree, scaled by the shrinkage factor $\gamma$.

$$\check{O}_i\ \ \check{O}_i + \gamma F_T(x) \tag{7}$$

 4. Residual update

$$R_i \leftarrow R_i - \check{O}_i \tag{8}$$

iii. The resulting boosted model is

$$\check{O}_i = \sum_{T=1}^{N}\gamma F_T(x) \tag{9}$$

Where, $\Omega(F_T) = \partial I + \dfrac{1}{2}\Delta\sum_{j=1}^{I}\omega_j^2$, $l$ represents a convex loss function, that quantifies the difference between the predicted ($\check{O}_i$) and the target value ($O_i$), $F_T$ is the Tth decision tree, $x_i$ denotes the ith sample, R represents the residual, N and I represent the tree and leaf count, $\partial\ and\ \Delta$ are coefficients of regularization.

### 3.4 Model Construction

The monthly runoff in the Lotschental catchment, located in Switzerland, has been predicted using various AI-based models, including XGBoost, RF, and a deep learning algorithm called LSTM. This study utilized runoff data spanning twenty years, from January 2002 to December 2021. The majority of the data, constituting 70% and covering the years 2002–2015, was employed for training the models and optimizing their calibrations (See Fig 5). The remaining data, approximately from 2016 to 2022, was reserved for testing the models' accuracies.

To identify the most suitable input data, partial autocorrelation was employed for selecting the best input lags. This method provides valuable insights into the time series properties, including stationarity, trend patterns, seasonality, and randomness [72]. The partial

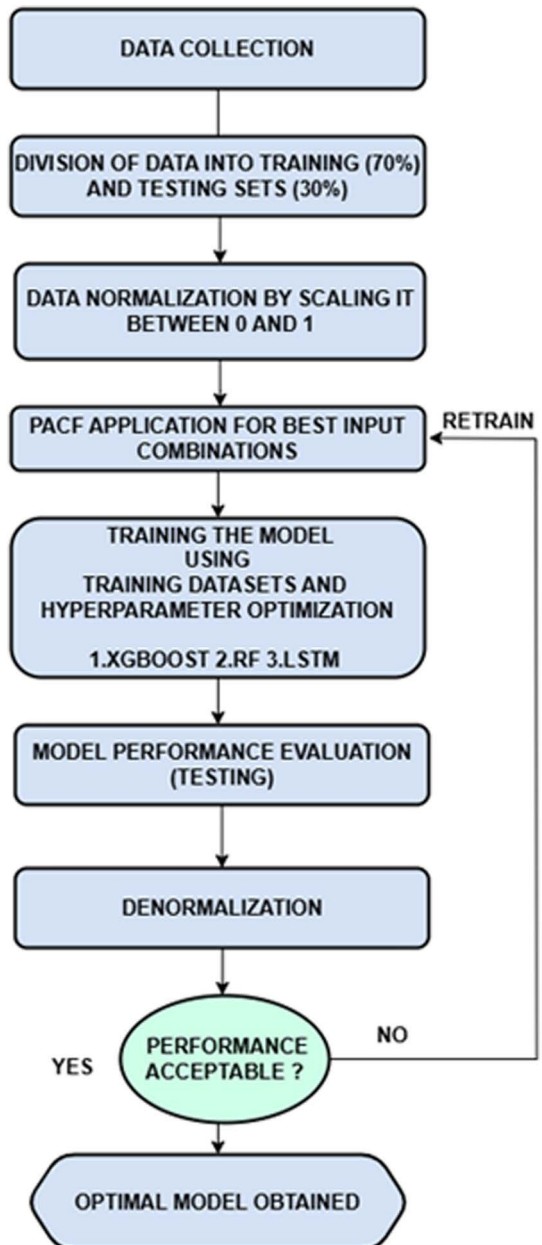

**Fig 5. Modeling Process.**

autocorrelation function (PACF) analysis, depicted in Fig 6 led to the establishment of three different input combinations, as illustrated in Table 2.

### 3.5 Statistical metrics

In this research, various statistical metrics commonly employed in the hydrological field were used to evaluate the forecasting reliability of each model [73]. These metrics include root mean square error (*RMSE*), mean absolute error (*MAE*), correlation coefficient and determination (*R*, and *R²*), Maximum absolute relative error (*erMAX*), Nash-Sutcliffe efficiency index (*NSE*), Willmot index (*d*), and uncertainty at a 95% confidence interval ($U_{95}$) [74]. Also, the

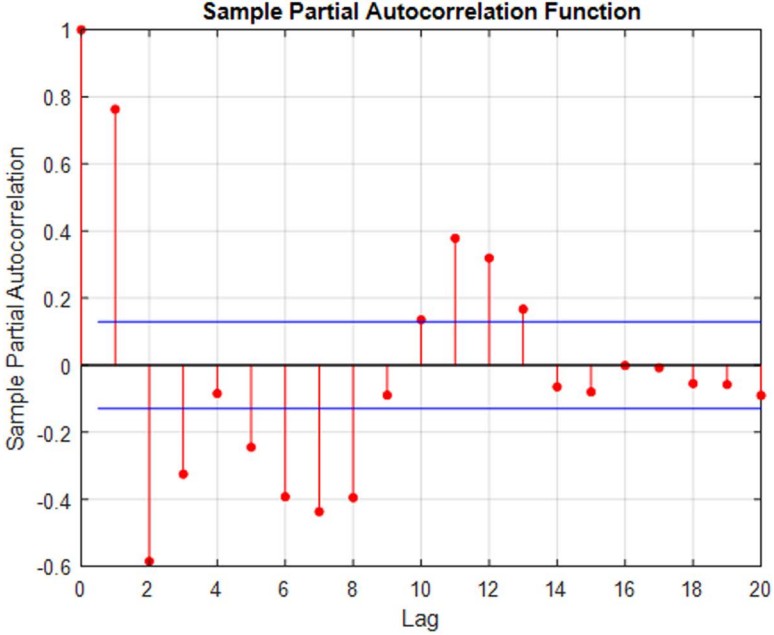

**Fig 6. Partial autocorrelation function for selecting the best lag runoff values.**

**Table 2. The input combinations adopted for forecasting runoff.**

| Combinations | Input and output |
|---|---|
| Comb1 | $Q_t = f\left( Q_{t-1}, Q_{t-2}, Q_{t-3} \right)$ |
| Comb2 | $Q_t = f\left( Q_{t-1}, Q_{t-2}, Q_{t-3}, Q_{t-5}, Q_{t-6}, Q_{t-7}, Q_{t-8} \right)$ |
| Comb3 | $Q_t = f\left( Q_{t-1}, Q_{t-2}, Q_{t-3}, Q_{t-5}, Q_{t-6}, Q_{t-7}, Q_{t-8}, Q_{t-11}, Q_{t-12}, Q_{t-13} \right)$ |

current research used a combined accuracy (*CA*) [75,76] which considered as efficient and global metrics, taking advantages of both error, and accuracy metrics. The mathematical equations for these metrics are provided in the following equations [77,78]:

$$RMSE = \sqrt{\frac{1}{n}\sum_{i=1}^{n}\left(Q_{obs_i} - Q_{pred_i}\right)^2} \qquad (10)$$

$$MAE = \frac{\sum_{i=1}^{n}\left|Q_{obs_i} - Q_{pred_i}\right|}{n} \qquad (11)$$

$$NSE = 1 - \frac{\sum_{i=1}^{n}\left|Q_{obs_i} - Q_{pred_i}\right|}{\sum_{i=1}^{n}\left|Q_{obs_i} - \overline{Q_{obs}}\right|} \qquad (12)$$

$$WI = 1 - \frac{\sum_{i=1}^{n}(Q_{obs_i} - Q_{pred_i})^2}{\sum_{i=1}^{n}\left(\left|Q_{pred_i} - \overline{Q}\right| + \left|Q_{obs_i} - \overline{Q_{obs}}\right|\right)^2} \qquad (13)$$

$$R^2 = 1 - \frac{\sum_{i=1}^{n}(Q_{obs_i} - Q_{pred_i})^2}{\sum_{i=1}^{n}\left(Q_{obs_i} - \overline{Q_{obs}}\right)^2} \tag{14}$$

$$R = \frac{\sum_{i=1}^{n}\left[\left(Q_{obs_i} - \overline{Q_{obs}}\right)\left(Q_{pred_i} - \overline{Q_{pred}}\right)\right]}{\sqrt{\sum_{i=1}^{n}\left(Q_{obs_i} - \overline{SSI_{obs}}\right)^2 \sum_{i=1}^{n}\left(Q_{pred_i} - \overline{Q_{pred}}\right)^2}} \tag{15}$$

$$U_{95} = \frac{1.96}{n}\sqrt{\sum_{i=1}^{n}\left(Q_{obs_i} - \overline{Q_{obs}}\right)^2 + \sum_{i=1}^{n}(Q_{obs_i} - Q_{pred_i})^2} \tag{16}$$

$$erMax = max\left(\frac{\left|Q_{obs_i} - Q_{pred_i}\right|}{Q_{obs_i}}\right) \tag{17}$$

$$CA = 0.33 \times \left(MAE + RMSE + \left(1 - R^2\right)\right) \tag{18}$$

Where for this study, $Q_{obs_i}$, $Q_{pred_i}$ are the observed runoff at $i^{th}$ data point and forecasted monthly runoff (Q), $n$ is the total data points. The means of observed and forecasted Q values are respectively, $\overline{Q_{obs}}$, $\overline{Q_{pred}}$.

### 3.6  Turning points in monthly runoff time series data

Turning points are essential in time series analysis as they indicate significant changes in the data's trend or direction. They mark transitions from one phase to another, highlighting shifts in behavior, growth, or decline within the dataset. These points provide critical insights into major events, changes in underlying processes, and the rise of new trends. Identifying turning points, such as peaks, troughs, cycles, and anomalies, is crucial for decision-making. Since time series data exhibit substantial autocorrelation (See Fig 6), forecasting models often achieve high accuracy ($R > 0.9$) [79] and low forecasting errors, according to metrics like *RMSE* and *MAE* [80]. However, special attention should be devoted to accuracy, specifically at turning points when evaluating forecasting models.

This study employed the Average Absolute Error of Turning Points (*AAETP*) to assess the accuracy of forecasting models in predicting turning points within a runoff dataset. The *AAETP* is a performance metric that can be used in time series analysis, focusing on the models' abilities to replicate significant shifts and turning points in the data. To identify turning points, the first and second derivatives of the original time series data were computed. The first derivative captures the rate of change or slope of the data, while the second derivative identifies the points where the sign of the first derivative changes, indicating turning points. By comparing the forecasted monthly runoff values with the actual observed values at the identified turning points, the average absolute difference was calculated independently for each model. For more details, please refer to [63] where the mathematical equation for *AAETP* can be found in Equation 18. The analysis of error magnitudes specifically at turning points provides valuable insights into the models' ability to accurately capture data shifts and transitions, thereby improving the assessment of their overall forecasting capabilities.

$$AAETP = \text{average}\left(\left|E_i\right|\right) \tag{18}$$

Where, $E_i$ represents the forecasting error for turning points. It is computed by subtracting the forecasted values of the critical points from their corresponding actual runoff values.

The theoretical optimum value for *AAETP* is zero, indicating that the model perfectly captures turning points. However, in practice, achieving a value of zero is very difficult. Therefore, it is essential to identify the minimum values that are closest to zero when comparing models. These values serve as indicators of the most effective models, facilitating a more reliable evaluation of how well they capture turning points

## 4. Results and discussion

### 4.1 Forecasting runoff using AI-based models

This study aimed to develop runoff forecasting models for the Lotschental catchment for one-month ahead timescale through three particular techniques Random Forest (RF), Long short-term Memory (LSTM), and XGBoost. In order to determine the optimal number of input lags, a backward selection strategy called Partial Autocorrelation Function (PACF) was applied. This resulted in three different sets of input variables. During the training stage, rigorous evaluation of each model employed several statistical measures such as Mean Absolute error (*MAE*), Root Mean Square Error (*RMSE*), Willmot index (*d*), Correlation Coefficient (*R*), Maximum absolute relative error (*erMAX*) and Nash-Sutcliffe efficiency index (*NSE*), and Combined accuracy(*CA*) to represent forecasting of models, error measurement and the degree of correlation and proportion of variance between the independent variables and the dependent variable respectively (refer to Table 3). These parameters played a key role in assessing the relative performance of the models in predicting runoff for Lotschental catchment

This study utilized three forecasting models (RF, LSTM, and XGBoost) to forecast runoff one month ahead for Lotschental catchment. The input lags were selected using PACF, resulting in three different combinations. The performance of each forecasting model during the training phase was evaluated using various statistical metrics (refer to Table 3).

The results demonstrated that all the models exhibited good forecasting accuracy, with the second and third combinations appearing to be more suitable, although the second combination shows a lower forecasting error. For example, XGBoost presented a higher *MAE* of 0.937 m³/sec in Comb1, whereas this value was reduced to 0.752 m³/sec and 0.776 m³/sec in Comb2 and Comb3, respectively. Similarly, for LSTM, the *MAE* values for the three combinations were 1.357 m³/sec, 0.784 m³/sec, and 0.821 m³/sec during the training phase. RF, on the other hand, exhibited slightly better performance than XGBoost across all three combinations, with lower *MAE* values of 0.727 m³/sec, 0.613 m³/sec, and 0.582 m³/sec.

**Table 3. Evaluating the performance of each model using different statistical metrics in the training phase.**

| Combinations | Models | RMSE m³/sec | MAE m³/sec | erMax | d | NSE | R | CA |
|---|---|---|---|---|---|---|---|---|
| Comb1 | XGBoost | 1.302 | 0.937 | 3.645 | 0.979 | 0.767 | 0.960 | 0.772 |
| | LSTM | 1.736 | 1.357 | 3.824 | 0.960 | 0.663 | 0.931 | 1.074 |
| | RF | 1.185 | 0.727 | 2.187 | 0.983 | 0.819 | 0.967 | 0.658 |
| Comb2 | XGBoost | 1.157 | 0.752 | 2.309 | 0.984 | 0.810 | 0.968 | 0.657 |
| | LSTM | 1.296 | 0.784 | 0.965 | 0.979 | 0.802 | 0.959 | 0.719 |
| | RF | 1.054 | 0.613 | 2.113 | 0.986 | 0.845 | 0.974 | 0.572 |
| Comb3 | XGBoost | 1.156 | 0.776 | 1.142 | 0.984 | 0.806 | 0.968 | 0.664 |
| | LSTM | 1.295 | 0.821 | 0.860 | 0.978 | 0.794 | 0.961 | 0.730 |
| | RF | 0.993 | 0.582 | 1.638 | 0.988 | 0.854 | 0.977 | 0.540 |

with lowest *MAE* and *CA* values of 0.727 m³/sec, 0.613 m³/sec, and 0.582 m³/sec, and 0.658, 0.572, and 0.540, respectively. However, XGBoost and RF showed similar accuracy parameters, such as *R*, during the training phase. It's important to note that the training phase alone is not crucial for selecting the best model since the models are trained with available input and output runoff data. The testing phase, where the model is only provided with input data, determines the generalization capacity and efficiency of the model.

The hyperparameters of the applied models are selected using a trial-and-error method, with the objective function being *RMSE*. The models are trained on calibration data, and hyperparameters are chosen based on their performance in reducing *RMSE*. For example, the LSTM model is very sensitive to hidden units and dropout values, with hidden units ranging from 15 to 35 and dropout rates from 0.09 to 0.2. In the RF model, sensitivity was noted regarding the number of trees, with the optimum found to be 18 and the minimum number of samples required at a leaf node set at 5. Finally, for the XGBoost model, the best learning rate was determined to be 0.15, while the maximum tree depth was set to 7. S2 Table presents the hyperparameters for all models. Notably, the hyperparameters for each forecasting model used to forecast monthly runoff are listed in the S Appendix (S2 Table).

Table 4 presents the performance of the forecasting models during the testing phase. Based on statistical parameters, the combination labeled Comb2, which involves seven previous runoff lags, exhibits the most favorable scenario for forecasting models. Within this particular scenario, XGBoost demonstrates superior performance compared to the other models, yielding the lowest values for *RMSE* (1.554 m³/sec), *MAE* (0.976 m³/sec), *erMax* (1.080), and *CA* (0.871), as well as the highest values for *d* (0.972), *NSE* (0.797), and *R* (0.956). Additionally, the XGBoost model achieves higher prediction accuracy by reducing *RMSE* by 14.57% and 17.03% compared to the LSTM and RF models, respectively. Overall, the statistical results from Tables 3 and 4 indicate that the XGBoost model outperformed the other models for all three experimental combinations. The model was the one to have the best performance metrics throughout, followed by the LSTM network and the Random Forest (RF) model, which presented relatively below-par performance. It is important to note that the RF model is prone to overfitting, as it demonstrates excellent forecasting performance during the training phase but performs less effectively during the testing phase. These findings underscore the efficiency of XGBoost in forecasting runoff and its superior generalization capacity when compared to LSTM and RF.

To comprehensively assess the predictive accuracy of the applied model, it is vital to verify its generalization capacity and accuracy in forecasting runoff across different seasons. Such evaluation provides further insights into how a model performs in each season, allowing us

**Table 4. Evaluating the performance of each model using different statistical metrics in the Testing phase.**

| Combinations | Models | *RMSE* m³/sec | *MAE* m³/sec | *erMax* | *d* | *NSE* | *R* | *CA* |
|---|---|---|---|---|---|---|---|---|
| Comb1 | XGBoost | 2.032 | 1.252 | 2.938 | 0.951 | 0.713 | 0.916 | 1.147 |
|  | LSTM | 2.426 | 1.560 | 4.359 | 0.927 | 0.642 | 0.876 | 1.405 |
|  | RF | 2.823 | 1.351 | 2.094 | 0.899 | 0.690 | 0.835 | 1.491 |
| Comb2 | XGBoost | 1.554 | 0.976 | 1.080 | 0.972 | 0.797 | 0.956 | 0.871 |
|  | LSTM | 1.819 | 1.065 | 1.479 | 0.962 | 0.757 | 0.945 | 0.996 |
|  | RF | 1.873 | 1.063 | 2.725 | 0.957 | 0.737 | 0.925 | 1.026 |
| Comb3 | XGBoost | 1.677 | 1.082 | 1.167 | 0.969 | 0.755 | 0.945 | 0.954 |
|  | LSTM | 1.769 | 1.107 | 1.704 | 0.961 | 0.750 | 0.942 | 0.995 |
|  | RF | 1.945 | 1.155 | 2.765 | 0.947 | 0.729 | 0.925 | 1.080 |

to understand the significant differences in model performance under varying conditions. According to Table 5, XGBoost is the best performing model, offering predicted values that closely align with those measured over the seasonal period. Notably, XGBoost delivers the highest forecasting accuracy during the summer months (July to September). Additionally, the other models demonstrate very good forecasting accuracy from July to September. XGBoost has the lowest overall relative percentage error (4.15%), followed by LSTM (6.16%), while RF generates the lowest accuracy (7.07%).

## 4.2  Graphical assessment

This chapter aims to visually demonstrate the effectiveness of each model in simulating the runoff time series data for each month. The line graph presented in Fig 7(a, b, c) illustrates that, in general, the models have successfully forecasted the runoff from May 2016 to approximately March 2019, capturing the overall pattern of the measured values. However, during this period, only the RF model exhibits relatively poorer forecast performance in terms of peak values (refer to Fig 6c).

From March 2019 to December 2021, the efficiency of the models varies significantly due to the presence of several peak points and substantial changes in the time series data. The XGBoost model demonstrates the closest fit to the measured values, with its forecasted values closely aligning with the measured values throughout the time series, particularly during peak points. Similarly, the LSTM model shows a good fit, with its forecasted values generally following the measured values, although there are some instances where slight deviations occur (refer to Fig 6b). On the other hand, the RF model exhibits a slightly wider range of forecasted values compared to the other two models, suggesting that it is less precise in its forecasts (refer to Fig 7c). According to the figure, the highest positive peak value is 16.887 m³/sec, which is simulated very well by XGBoost and LSTM, with predicted values of 15.8861 m³/sec and 15.4421 m³/sec, respectively. However, RF exhibits lower prediction performance for these peak points, with a value of 15.3793 m³/sec. With regard to the lowest runoff point, which is measured at 0.444 m³/sec, only XGBoost simulates this point with reasonable accuracy at 0.6851 m³/sec, while the other models generate values ranging from 1 m³/sec to 1.68 m³/sec. The accurate simulation of both maximum and minimum runoff points demonstrates the adaptability of the XGBoost model in providing reliable forecasts under varying hydrological conditions, including both drought and flood events.

The scatter plots in Fig 6 a1, b1, and c1 illustrate the relationship between measured runoff and the forecasted values from the models in Comb2. The scatter plot for XGBoost (Fig 7 a1) demonstrates a strong positive correlation with a tight clustering of data points around the diagonal line, indicating accurate predictions ($R^2 = 0.901$). The LSTM plot (Fig 6b1) shows slightly more spread data points compared to XGBoost but less than RF, suggesting some errors in predictions ($R^2 = 0.869$). Similarly, the RF plot ((Fig 6 c1) exhibits a wider spread of data points, indicating deviations from the actual values ($R^2 = 0.861$). Overall, The XGBoost model is the best-performing model, followed by LSTM and then RF.

**Table 5.  Performance of predictive models for seasonal runoff forecasting.**

| Seasons | Observed (m³/sec) | XGBoost (m³/sec) | LSTM (m³/sec) | RF (m³/sec) |
|---|---|---|---|---|
| April to June | 6.8620 | 6.5370 | 6.3467 | 6.1522 |
| July to September | 10.3389 | 10.0101 | 9.9042 | 9.5184 |
| October to November | 1.5456 | 1.4215 | 1.3412 | 1.7501 |
| Overall | 6.2488 | 5.9895 | 5.8640 | 5.8069 |
| Overall relative percentage error % | | 4.15 | 6.16 | 7.07 |

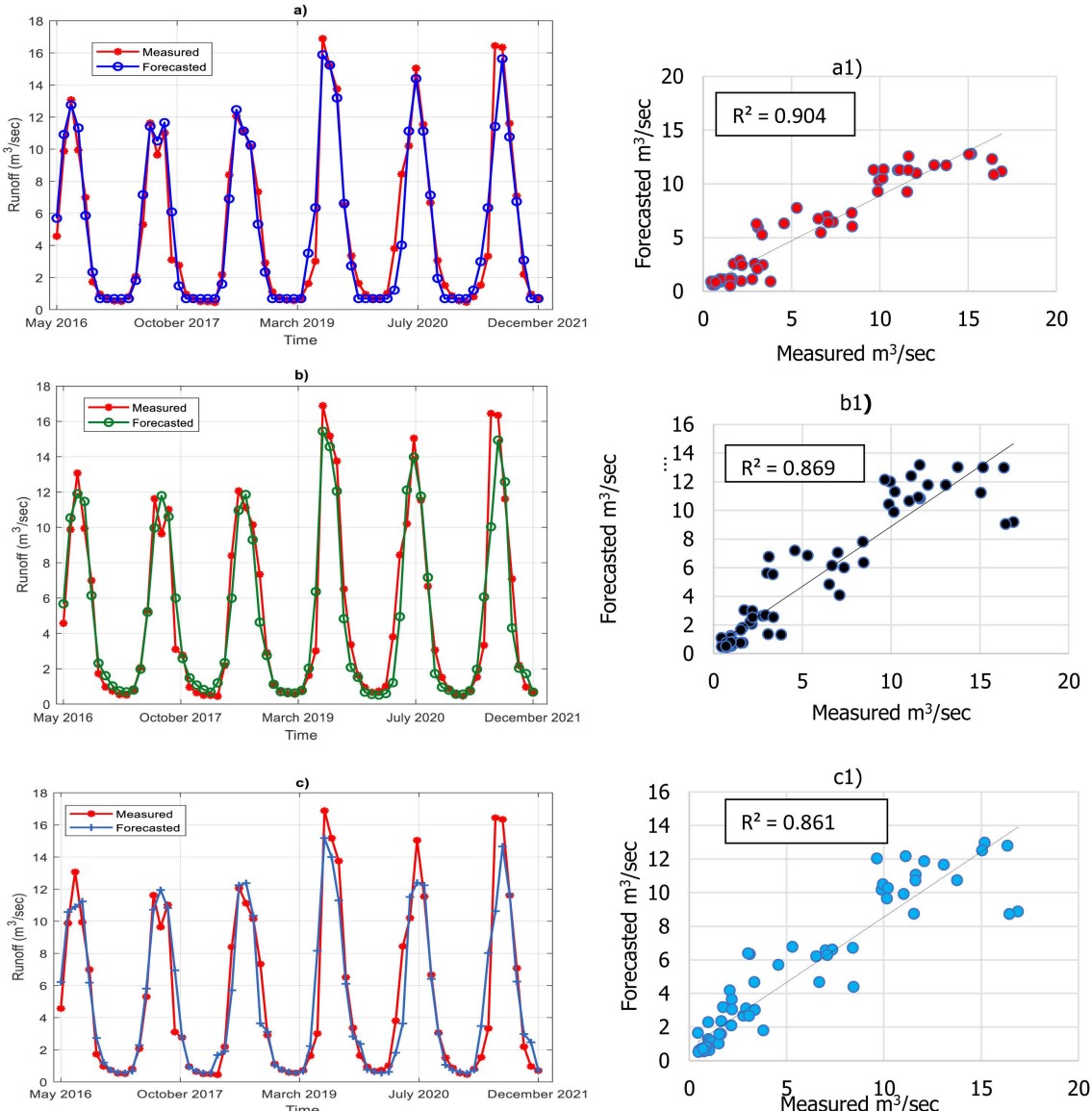

**Fig 7. Line graph and Scatter plots comparing measured monthly runoff and forecasted values.** Panels a) XGBoost (Line Graph), a1) XGBoost (scatter Plot), b) LSTM (Line Graph), b1) LSTM (Scatter Plot), c) RF (Line Graph), c1) RF (Scatter Plot).

Based on the results displayed in Fig 8, which presents the seasonal performance of each model, it is evident that all models exhibit varying accuracy across seasons, reflecting fluctuations in the runoff pattern over time. Overall, XGBoost demonstrates strong performance throughout all three seasonal periods, yielding the lowest absolute residuals of 0.325 m³/sec, 0.3288 m³/sec, and 0.1241 m³/sec, respectively. Notably, during the period from April to June, XGBoost consistently outperforms the other models. Overall, XGBoost consistently demonstrates the highest accuracy in modeling seasonal runoff patterns and effectively captures variations across various time periods.

A violin plot represents the distribution of numerical data across one or more groups through density curves. The width of each curve indicates the relative frequency of data points within specific intervals. As depicted in Fig 9, violin plot has been employed to facilitate a

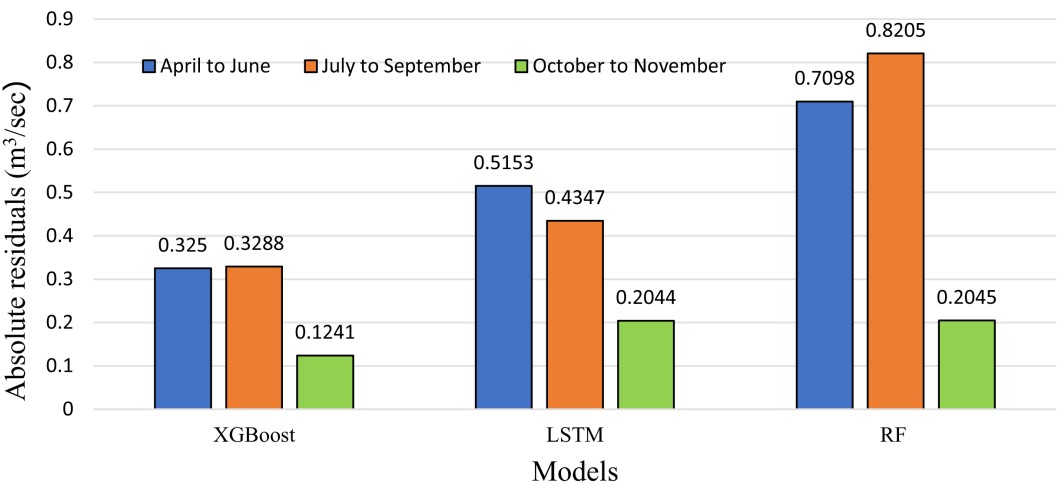

**Fig 8. Absolute residual errors showing seasonal variation in model performance.**

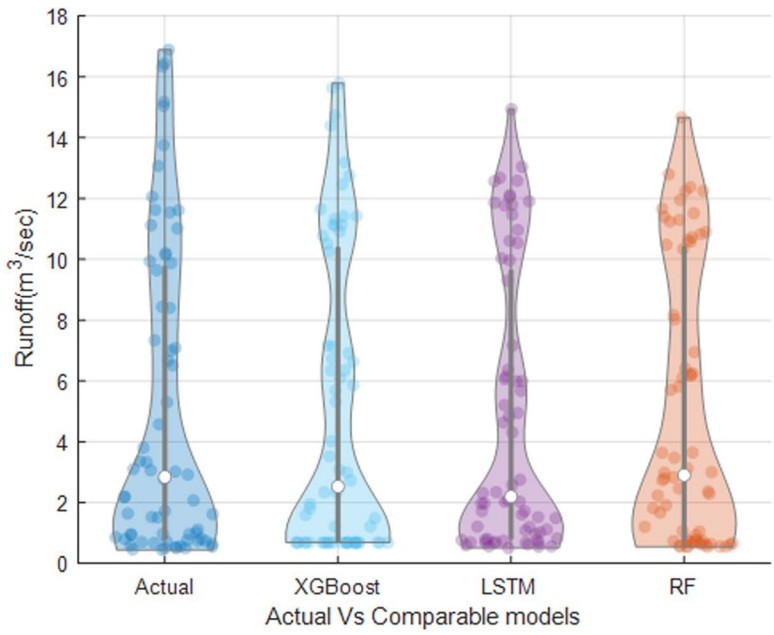

**Fig 9. Violin plot illustrating the comparison between observed and simulated Runoff.**

comprehensive comparison between the measured runoff and the forecasted runoff obtained from the applied models. In general, all models capture the overall trend and distribution of the runoff data. However, variations among the models become evident, particularly in the peak points. Notably, XGBoost demonstrates favorable forecasting performance for the peak runoff points when compared to the LSTM and RF models.

Moreover, the Taylor diagram has been used as a valuable tool to identify the most efficient model. These plots are mathematical constructs that graphically illustrate the relative accuracy of various model representations of a system, process, or phenomenon. The diagrams facilitate comparative analyses and provide a comprehensive visualization by summarizing the

correlation, standard deviation ratio, and root mean square difference between the reference dataset (measured runoff) and multiple model simulations in a single plot. Fig 10 demonstrates that XGBoost exhibits the closest proximity to the reference points (measured runoff), followed by the LSTM and RF models. Finally, uncertainty analysis, specifically in terms of $U_{95}$, was performed to determine which model produces the least uncertainty. According to Fig 11, XGBoost demonstrates the lowest $U_{95}$ value of 1.231, which is lower than LSTM with

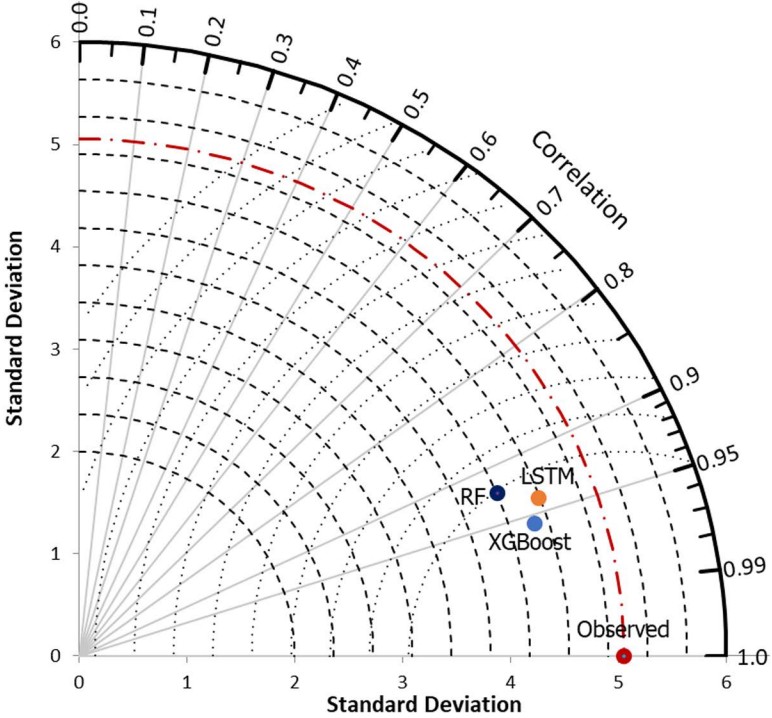

**Fig 10. Taylor diagram for visual comparisons among comparable models.**

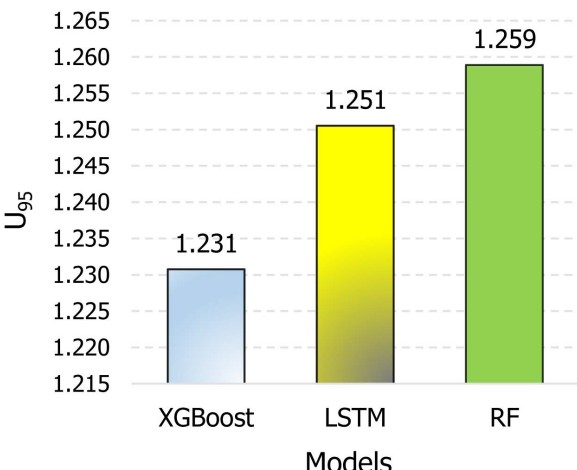

**Fig 11. Evaluating model performance in the testing phase: a comparative analysis based on *U95* indicator.**

1.251 and RF with 1.259. Overall, the analysis of the Taylor plot, violin plot, and $U_{95}$ indicates that XGBoost is the most effective model for predicting monthly runoff in the study area, demonstrating the lowest level of uncertainty.

### 4.3  Analyzing forecasting models with turning point detection

This section of the paper focused on the assessment focused on the applied model's efficiency in detecting runoff turning points. Fig 12a visually depicts how the algorithm identifies turning points in the time series runoff data. The graph illustrates a cyclical pattern in the time series runoff data, with clearly defined peaks and valleys. The turning points denoted on the graph highlight substantial changes in the data. It is important to note that the graph enables an assessment of the accuracy of the applied models in predicting these critical turning points.

The bar chart in Fig 12b showcases the accuracy of each applied model in forecasting the identified turning points. The comparison of *AAETP* values reveals that XGboost performs the best among the models, with an *AAETP* of 1.579 m³/sec. In contrast, the LSTM model has an *AAETP* of 1.924 m³/sec, and the RF model has an *AAETP* of 2.119 m³/sec. This indicates that XGboost outperforms LSTM and RF in accurately forecasting the turning points of runoff. Specifically, the accuracy of turning point forecast improves by approximately 21% and 34% compared to LSTM and RF models, respectively.

### 4.4  Discussion

The current study investigated the effectiveness of three different models, namely LSTM, RF, and XGBoost, in forecasting monthly runoff for the Lotschental catchment in Switzerland. Among these models, XGBoost demonstrated superior accuracy, achieving a forecasting

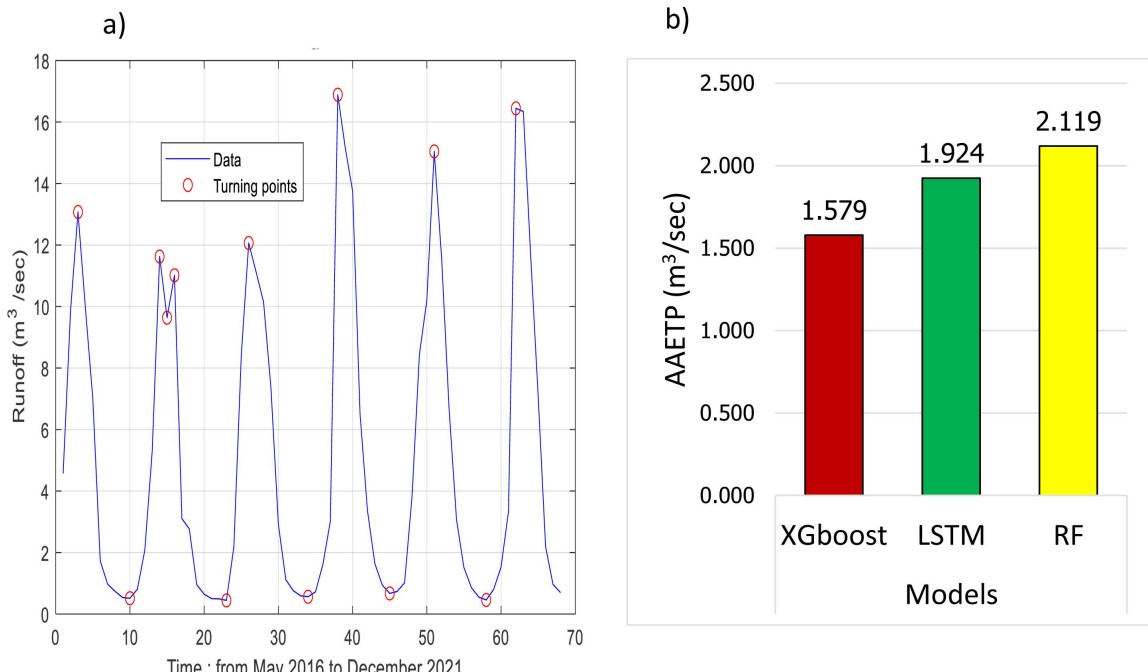

**Fig 12.  Monthly runoff turning point detection:**  a) illustration of the algorithm's capability in detecting turning points in the runoff time series data. b) calculated results of *AAETP*.

accuracy of $R = 0.956$. These findings contribute to advancing the accuracy of previous studies in this field. In a similar vein, some researchers [81] have employed the hybrid model ELM-GWO, which combines the extreme learning machine (ELM) with the grey wolf optimizer (GWO), to forecast runoff in the Three Gorges region of China. During the testing phase, the hybrid model (ELM-GWO) exhibited good accuracy, achieving an $R$-value of 0.9228. Another study explored the application of the Emotional Artificial Neural Network (EANN) for monthly runoff forecasting in the Kallada River basin, located in Kerala [82]. The findings of this research demonstrated that the EANN model yielded satisfactory results, achieving a forecasting accuracy with an $R$-value of 0.877. Furthermore, machine learning models have also been utilized in the semi-arid Central Andes of Argentina to predict the runoff [83]. The study concluded that support vector regression (SVR) demonstrated effective forecasting of the runoff, achieving $R^2$ values ranging from 0.75 to 0.89 during the testing phase. Moreover, the XGBoost model developed in this study demonstrates good accuracy compared to similar studies conducted to forecast runoff in glacierized catchments. For example, some researchers conducted a study forecasting runoff using a conceptual hydrological model called Glacial Snow Melt (GSM) and SVR based on several hydrological and meteorological parameters [84]. The results showed that the GSM achieved a good accuracy with an $R^2$ of 0.77, while the standalone SVR reached an $R^2$ of 0.83. Similarly, another study developed a conceptual hydrological model for modeling runoff in glacierized alpine areas and found that the model provided good accuracy, with $R^2$ values ranging from 0.74 to 0.89 [85]. Among the models examined in the literature, XGBoost demonstrated superior forecasting accuracy compared to others in forecasting monthly runoff for the Lotschental catchment, highlighting its effectiveness in this context. Furthermore, it is important to highlight that the forecasting models developed in previous research did not include turning point analysis. However, this research demonstrates that such a metric is highly significant in evaluating and comparing models.

As the XGBoost model in this study provides good forecasting accuracy for modeling monthly runoff compared to previous works, it may also be suitable for forecasting runoff over other time scales, such as daily and weekly. In shorter time scales (daily and weekly), the autocorrelation in time series data is typically higher than that in monthly scales, which may increase the opportunity for achieving better results. Additionally, in shorter time scales, the number of data points is greater than in the monthly scale, which may help models to be trained effectively with sufficient data.

In this work, a deep learning model called LSTM is used for modeling time series data because of its ability to remember past events and capture patterns over time. While LSTM addresses the vanishing gradient problem faced by traditional models like recurrent neural network (RNNs) [86], it did not outperform XBGoost in modeling runoff. This is largely because LSTM requires a significant amount of data to train effectively and may need additional features to capture complex temporal dependencies in the runoff data. The results of the current study indicate that deep learning model (LSTM) were not superior to machine learning models (XGBoost). Thus, the findings of this research align with another study [87] on runoff prediction that found deep learning models gave an acceptable result but less accurate than their machine learning counterparts.

The selection of training and testing samples for model training and validation is a crucial step in developing predictive models. In this study, runoff measurements from 2002 to 2015 are utilized for model development, while data from 2016 to 2021 serve to assess the accuracy of the model. Ensuring balanced data and minimizing bias between the training and testing sets play a significant role in creating a robust model with strong generalization capacity [88]. Fig 13 summarizes the key statistical metrics for both datasets. Notably, there is a strong similarity in the training characteristics. For instance, the minimum, maximum, average, and

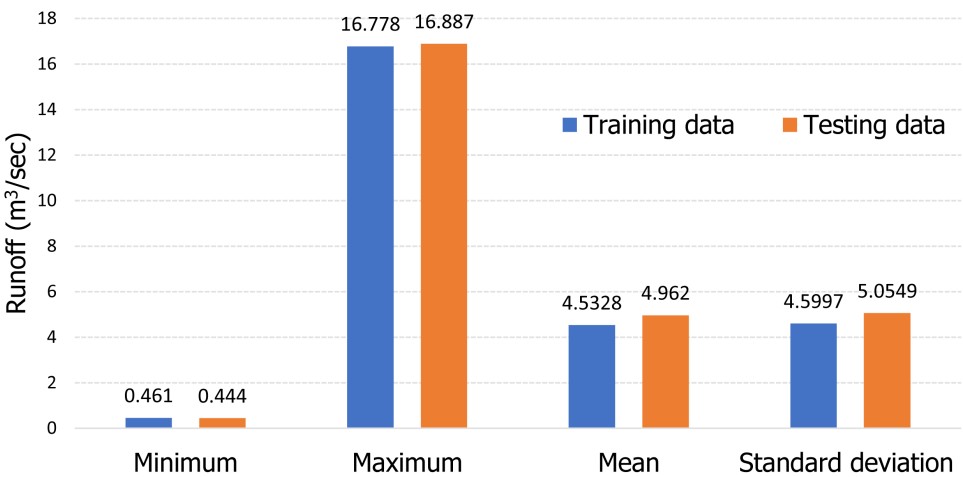

**Fig 13. Key statistical metrics of training and testing datasets.**

standard deviation values for the training and testing datasets are as follows: 0.461 m³/sec vs. 0.444 m³/sec, 16.778 m³/sec vs. 16.887 m³/sec, 4.5328 m³/sec vs. 4.962 m³/sec, and 4.5997 m³/sec vs. 5.0549 m³/sec. These statistics indicate that both datasets are homogeneous and well-balanced, further supporting the validity of the model.

The choice of input lag combinations plays a critical role in time series forecasting (e.g., runoff) as it can significantly affect the performance of forecasting models such as LSTM, XGBoost, and RF. Also, using various combinations of input lags helps to evaluate their impact on the performance of prediction models. In this work, three combinations are used based on PACF. The first combination involves only three input variables, while the second includes seven lags, and the last one involves a large number of variables, up to ten. The performance of the models varies from one combination to another. The models attempt to capture the overall temporal dependencies in the testing data and require optimum input choices. It can be seen that none of the models provided good prediction accuracy using the first combination due to the lack of significant information needed to effectively capture the patterns of the time series data. In this case, XGBoost achieved the lowest forecasting error with *MAE* of 1.252 m³/sec, followed by RF (1.351 m³/sec) and LSTM (1.560 m³/sec). Regarding the second input combination, all models presented significant improvement, with XGBoost providing an *MAE* of 0.976 m³/sec, followed by RF (1.063 m³/sec) and LSTM (1.065 m³/sec). However, the forecasting accuracy of all the models deteriorated with the third input combination, with the *MAE* for XGBoost, RF, and LSTM being higher than those in the second combination. The main reason may be related to the introduction of a large number of features, which could lead to redundancy in the data and significantly affect the generalization of the models, complicating the training process. Once the input combination contained fewer variables, as seen in the first scenario, the models exhibited lower prediction capacity because the available inputs did not contain significant information that could be mapped by the models to mimic the patterns of the time series data and effectively capture the temporal dependencies in the runoff.

The melting of snow and glaciers is increasingly important for runoff generation due to the impacts of climate change and global warming. Enhancing the accuracy of runoff forecasting models offers tangible benefits for managing water resources in cold regions affected by such environmental changes. This is particularly crucial for the effective allocation and conservation of water resources. In our study, we found that traditional models had limited

effectiveness in forecasting peak flows, which are critical for water management and flood mitigation in climate-affected regions. Conversely, the XGBoost model demonstrated notable improvements, producing the most accurate runoff simulations. The model's performance metrics, including *RMSE* of 1.554 m³/s, *MAE* of 0.976 m³/s, $U_{95}$ of 1.231, $R^2$ of 0.91, and *d* of 0.972, reflect its proficiency. Besides, the model exhibited superior performance in forecasting runoff turning points, achieving the lowest *AAAET* (1.579 m³/sec) among the compared models. These results underscore the potential of XGBoost models in areas with glacierized catchments, which are particularly vulnerable to rapid climate change and glacial melt. The study offers evidence that the proposed model adeptly handles the complex, nonlinear interplay between historical and projected runoff data, thus enabling more reliable water resource management in these challenging environments.

Furthermore, as shown in Fig 14, the model demonstrates excellent monthly runoff performance across nearly all months, generating minimal errors as indicated by the *RMSE* values. The lowest forecasting errors occurred in January and December, with *RMSE* values of 0.068 m³/sec and 0.124 m³/sec, respectively. In contrast, the highest forecasting error was recorded in June, with an *RMSE* of 3.354 m³/sec. The increase in forecasting error can be attributed to the instability and significant variability of the flow during this month. As shown in Fig 2, this period corresponds to the highest runoff into the basin, which leads to greater discrepancies in the model's predictions and subsequently higher forecasting errors.

## 5. Conclusion

The current study provides a comprehensive comparison between LSTM, XGBoost, and RF models for forecasting mean monthly runoff one month in advance. This study was conducted in the Lötschental catchment, Switzerland, which is classified as a glacierized catchment. The glacierized catchments are highly sensitive to climate change, where surface runoff undergoes dynamic hydrological processes influenced by climatic conditions and the formation and melting of ice. Historical data spanning from January 2002 to December 2021 was utilized, with input lag values determined using PACF and supplied to the runoff forecasting models. The data was divided into two phases: a training phase, which included runoff data from 2002 to 2015, and a testing phase, which used data from 2016 to 2021. The results

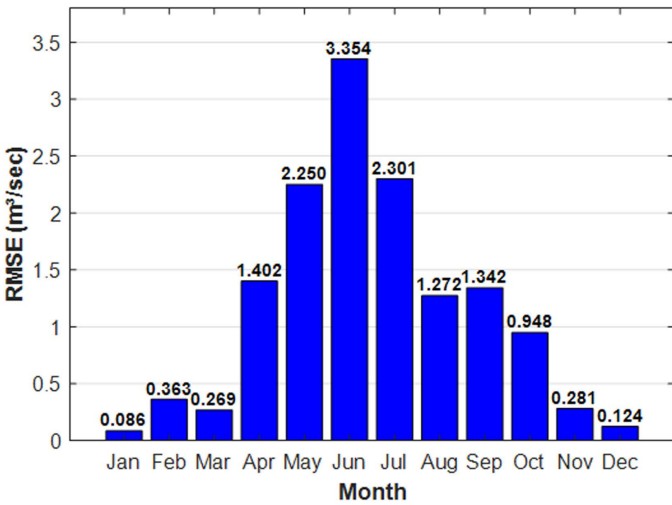

**Fig 14. Monthly *RMSE* values.**

indicated that XGBoost provided superior forecasting results compared to the other models, enhancing forecasting accuracy by reducing *RMSE* by 14.57% and 17.03% compared to the LSTM and RF models, respectively.

The current study also analyzes the capacity of the applied models to accurately forecast turning points in runoff data. Since the time series data of monthly runoff records contain significant autocorrelation, most AI models tend to provide in general good forecasts according to classical statistical metrics. Therefore, a new statistical criterion has been introduced in this research called *AAETP*. The quantitative results indicate that the XGBoost model was the most effective in accurately identifying turning points in the runoff time series data, outperforming the LSTM and RF models by approximately 21% and 34%, respectively. The effectiveness of the XGBoost model in capturing the complex runoff dynamics of glacierized catchments may have practical implications for water resource management decision-making. Also, its accurate forecasts may provide decision-makers with valuable insights to optimize strategies, allocate resources, and promote sustainable practices in response to changing hydrological conditions.

However, the proposed study still has some major constraints. Models' limitation to predict peak values of monthly runoff concentrations with high accuracy will be improved by using data decomposition techniques such as reliable wavelet transform techniques. Future studies could also investigate the potential of incorporating ensemble and hybrid models, such as CNN-LSTM (combining Convolutional Neural Network with LSTM), ELM-MHA (combining metaheuristic algorithms with ELM) and other hybrid techniques for predicting monthly runoff rates. Also, it is recommended to incorporate longer runoff and climate datasets from observed stations or satellites to enhance runoff forecasting and better understand the impacts of climate change.

## Supporting information

**S1 Table. Monthly runoff data that is used in this study.**
(DOCX)

**S2 Table. Hyperparameters for the applied models.**
(DOCX)

## Acknowledgment

The authors would like to thank University of Anbar for supporting this research.

## Author contributions

**Conceptualization:** Mohammed Majeed Hameed, Adil Masood, Aadil hamid.

**Data curation:** Ahmed Elbeltagi, Ali Salem.

**Formal analysis:** Mohammed Majeed Hameed, Adil Masood, Ahmed Elbeltagi, Ali Salem.

**Funding acquisition:** Ali Salem.

**Methodology:** Mohammed Majeed Hameed, Adil Masood.

**Resources:** Ahmed Elbeltagi, Ali Salem.

**Software:** Mohammed Majeed Hameed, Adil Masood.

**Visualization:** Mohammed Majeed Hameed, Adil Masood.

**Writing – original draft:** Mohammed Majeed Hameed, Aadil hamid, Siti Fatin Mohd Razali.

**Writing – review & editing:** Mohammed Majeed Hameed, Adil Masood, Ahmed Elbeltagi, Siti Fatin Mohd Razali, Ali Salem.

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
