## [Decision Letter · Decision Letter 0]

19 Oct 2024

PONE-D-24-40156Forecasting Monthly Runoff in a Glacierized Catchment: A Comparison of Extreme Gradient Boosting (XGBoost) and Deep Learning Models.PLOS ONE

Dear Dr. Salem,

Thank you for submitting your manuscript to PLOS ONE. After careful consideration, we feel that it has merit but does not fully meet PLOS ONE’s publication criteria as it currently stands. Therefore, we invite you to submit a revised version of the manuscript that addresses the points raised during the review process.

We look forward to receiving your revised manuscript.

Kind regards,

Shicheng Li

Academic Editor

PLOS ONE

Journal Requirements:

Reviewers' comments:

Reviewer's Responses to Questions

**Comments to the Author**

1. Is the manuscript technically sound, and do the data support the conclusions?

Reviewer #1: Yes

Reviewer #2: Yes

Reviewer #3: Partly

2. Has the statistical analysis been performed appropriately and rigorously? 

Reviewer #1: Yes

Reviewer #2: Yes

Reviewer #3: Yes

3. Have the authors made all data underlying the findings in their manuscript fully available?

Reviewer #1: Yes

Reviewer #2: Yes

Reviewer #3: Yes

4. Is the manuscript presented in an intelligible fashion and written in standard English?

Reviewer #1: Yes

Reviewer #2: Yes

Reviewer #3: Yes

5. Review Comments to the Author

Reviewer #1: In this work, the Extreme Gradient Boosting (XGBoost), random forest (RF) and long-short term memory (LSTM) model was used to forecast monthly runoff for one month ahead of time in the Lotschental watershed in Switzerland. The work is well organized. however, with the following minor corrections, it is acceptable:

SPECIFIC RECOMMENDATIONS

ABSTRACT

• The abstract could briefly explain why these methods were chosen or how they contribute to the research.

• The results could be more specific, including quantitative outcomes or more detailed

• At the end of the Abstract, it is necessary to point out the global importance of this study for society. This would arouse greater interest from Journal readers.

INTRODUCTION SECTION

• The introduction effectively covers the necessary elements but could be improved by providing a more detailed summary of methods, emphasizing the scientific gap, and enhancing the problem statement with a broader context.

• If possible, reinforce the introduction with up-to-date references (2022-2024) below. Because it is important to include the most recent articles.

1. Improving the accuracy of rainfall-runoff relationship estimation using signal processing techniques, bio-inspired swarm intelligence and artificial intelligence algorithms

2. Monthly streamflow prediction in Amasya, Türkiye, using an integrated approach of a feedforward backpropagation neural network and discrete wavelet transform

3. Improving the accuracy of rainfall-runoff relationship estimation using signal processing techniques, bio-inspired swarm intelligence and artificial intelligence algorithms

4. Application of empirical mode decomposition, particle swarm optimization, and support vector machine methods to predict stream flows

5. Simulation of the potential impacts of projected climate and land use change on runoff under CMIP6 scenarios

MATERIALS & METHOD

METHODOLOGY

• Consider adding brief explanations on why specific methods were chosen.

RESULTS

• Improve the logical flow and ensure that each finding is clearly explained.

• Please give applied model parameters.

CONCLUSIONS

• Clearly stating the most significant findings, with specific reference to the data.

• Offer more detailed and actionable recommendations based on the study's results.

• Highlight the study's contribution to existing research and its potential implications for theory and practice.

• Ensure that each paragraph is concise and directly contributes to the discussion.

Reviewer #2: This paper proposes Extreme Gradient Boosting (XGBoost) and Deep Learning Models to predict the Monthly Runoff in a Glacierized Catchment. However, the quality of the paper should be improved a bit. Specific comments are as follows.

1. The graphical abstract/flow diagram is missing, it must be added.

2. The novelty statement and research gap should be highlights in innovative aspect of this study, specifically in abstract and Introduction.

3. The shortcomings of existing research should be summarized, and the innovation of this paper should be highlighted.

4. As described in the paper, Extreme Gradient Boosting (XGBoost) and Deep Learning Models has been widely applied in the various fields, such as river stage forecasting (https://doi.org/10.1007/s13201-024-02103-8;
https://doi.org/10.1016/j.heliyon.2023.e16290), Drought Forecasting (https://doi.org/10.3390/w15040765) and long-term rainfall prediction (https://doi.org/10.1007/s00024-022-03189-4), Pan Evaporation (https://doi.org/10.1007/s00024-023-03426-4;
https://doi.org/10.1007/s13201-022-01846-6). However, several aspects in health monitoring, such as the reason for the adoption of Extreme Gradient Boosting (XGBoost) and Deep Learning Models in this paper is not clearly explained, it need to be appropriately supplemented.

5. The algorithms of machine learning models should be simplified, and the innovation of this paper should be highlighted. Readers are more concerned about the advantages of the deep learning model proposed in this study in forecasting.

6. Line no 62, " Several studies [11–13]" the citation must be at the end of sentence.

7. Lin No 79, the referencing and citation must be according to the journal style. for instant, water table [28] and river stages (Choi et al., 2020), both type of citations are there. it should be check throughout the manuscript.

8. Due to the extensive use of abbreviations, it is essential to include a list of acronyms for clarity and ease of understanding.

9. More datasets/no of stations should be provided to demonstrate the generalization ability and robustness of AI models.

10. What is new in this manuscript that was not there in the previous study? And what are the differences between this study and the old study? The limitations should be differentiated in discussion.

11. Future study must be stated.

Reviewer #3: The article titled "Forecasting Monthly Runoff in a Glacierized Catchment: A Comparison of Extreme Gradient Boosting (XGBoost) and Deep Learning Models" evaluates the effectiveness of XGBoost, LSTM, and Random Forest RF models in forecasting monthly runoff in the Lotschental catchment, Switzerland. This comparative analysis is beneficial for identifying the best model for monthly runoff forecasting under complex hydrological conditions, especially in glacierized catchments.

The article is interesting, but the following issues should be addressed.

Specific Comments and Suggestions:

1. Consider adding in Abstract a sentence highlighting the novelty of the study, such as the use of turning point analysis or the comparison of multiple machine learning models in this specific context.

2. In Introduction the literature review could be further strengthened by discussing recent advancements in machine learning and their potential applications in hydrological modeling.

The research questions could be explicitly stated at the end of the Introduction to provide a clear focus for the study.

3. Data Sources: Clearly indicate the specific sources of meteorological and hydrometric data used in the study.

4. Additional questions for consideration In the "Case Study Location" section:

- have there been any significant changes in land cover or vegetation patterns within the catchment in recent decades?

- are there any known geological faults or other factors that could influence the stability of the slopes and the risk of landslides?

- how have past flooding events affected the local communities and infrastructure?

5. Limited data temporal scope.

Although the study spans nearly two decades (2002-2021), integrating longer historical data could provide a more comprehensive understanding of long-term trends and improve the models' generalization capacity. Then the models could capture rare but critical climate events that affect runoff.

Why weren't earlier data and longer measurement series used?

6. Consider adding a more detailed explanation of the LSTM architecture

7. Discuss the process of hyperparameter tuning for each model, as this can significantly impact performance.

8. How did the choice of input lags affect the performance of the models?

9. Were any other machine learning models considered for comparison?

10. Were there any significant differences in model performance during different seasons or hydrological conditions?

11. How did the models perform in forecasting extreme events, such as floods or droughts?

12. While the study compares three individual models, it doesn’t explore hybrid modeling approaches. Combining the strengths of XGBoost and LSTM, for instance, could yield a model that balances the interpretability and robustness of XGBoost with the sequential learning advantages of LSTM.

The authors should refer to the issue of potential for hybrid model.

13. Although LSTM is a robust model for time series data, other advanced deep learning architectures, could further improve forecasting accuracy by capturing long-term dependencies more effectively. The authors should address this probleme.

14. The sensitivity of the AETP metric should be assessed to different thresholds for defining turning points. This can help to evaluate the robustness of the results.

15. Including climate projections could significantly enhance the model's utility for long-term water resource planning in glacierized regions that are highly sensitive to global warming.

16. Are there any specific regions or time periods where one model consistently outperforms the others?

17. To improve the "Discussion" section, it would be necessary to specify"

- how do the findings of this study compare to other research on runoff forecasting in glacierized catchments located in different regions?

- are there any potential biases or uncertainties associated with the use of historical data for model training and validation?

- could the XGBoost model be adapted or improved to forecast runoff at different time scales (e.g., daily, weekly)?

Incorporating longer data records, exploring hybrid models, and integrating climate change scenarios could further enhance the model's applicability in hydrological forecasting.

6. PLOS authors have the option to publish the peer review history of their article (what does this mean? ). If published, this will include your full peer review and any attached files.

**Do you want your identity to be public for this peer review?** For information about this choice, including consent withdrawal, please see our Privacy Policy .

Reviewer #1: No

Reviewer #2: No

Reviewer #3: No

---

## [Author Response · Author response to Decision Letter 1]

8 Feb 2025

Dear Editor,

We sincerely appreciate the opportunity to submit the revised version of our manuscript entitled “Forecasting Monthly Runoff in a Glacierized Catchment: A Comparison of Extreme Gradient Boosting (XGBoost) and Deep Learning Models.” to PLOS ONE.

We would also like to express our gratitude to the esteemed reviewers for their valuable and constructive comments. We have worked diligently to ensure that all authors have collaboratively addressed the comments raised by the reviewers and have made the necessary revisions to enhance the manuscript. Additionally, we would like to confirm that we have responded to the editorial comments, including the copyright concerns regarding Figure 1. We utilized a map sourced from open access source (please see Figure 1 in the main paper), and the relevant data availability details are incorporated within the manuscript and provided in the Appendix (please see the Suplimantary_file). Furthermore, our manuscript specifies that the models were developed using Python, which is open-access and freely available.

Below, we provide a point-by-point response to the comments raised by the reviewers:

Reviewer #1:

In this work, the Extreme Gradient Boosting (XGBoost), random forest (RF) and long-short term memory (LSTM) model was used to forecast monthly runoff for one month ahead of time in the Lotschental watershed in Switzerland. The work is well organized. however, with the following minor corrections, it is acceptable:

SPECIFIC RECOMMENDATIONS

ABSTRACT

- The abstract could briefly explain why these methods were chosen or how they contribute to the research.

Reply: We have explained in the abstract the reasons for selecting the applied models, as per your suggestion.

- The results could be more specific, including quantitative outcomes or more detailed.

Reply: We have incorporated additional quantitative results and details to make our findings more specific, as suggested.

- At the end of the Abstract, it is necessary to point out the global importance of this study for society. This would arouse greater interest from Journal readers.

Reply: We have emphasized the global significance of this study for society, as recommended.

INTRODUCTION SECTION

• The introduction effectively covers the necessary elements but could be improved by providing a more detailed summary of methods, emphasizing the scientific gap, and enhancing the problem statement with a broader context.

Reply: We have enhanced our introduction by including a summary of the applied models and highlighting the existing scientific gaps, as recommended. Please see pages 5 (lines 102 to 110) and 6-7 (lines 133 to 146).

• If possible, reinforce the introduction with up-to-date references (2022-2024) below. Because it is important to include the most recent articles.

1. Improving the accuracy of rainfall-runoff relationship estimation using signal processing techniques, bio-inspired swarm intelligence and artificial intelligence algorithms

2. Monthly streamflow prediction in Amasya, Türkiye, using an integrated approach of a feedforward backpropagation neural network and discrete wavelet transform

3. Improving the accuracy of rainfall-runoff relationship estimation using signal processing techniques, bio-inspired swarm intelligence and artificial intelligence algorithms

4. Application of empirical mode decomposition, particle swarm optimization, and support vector machine methods to predict stream flows

5. Simulation of the potential impacts of projected climate and land use change on runoff under CMIP6 scenarios

Reply: We would like to confirm that we have improved our paper by incorporating new references, as suggested. Below is a list of some additional references included in the paper:

1. Al-Kubaisi MHD. Surface Runoff Estimation in Kubaisa Watershed Using SWAT, Western Desert, Iraq. Iraqi Geological Journal. 2024;57: 286–297. doi:10.46717/igj.57.2B.19ms-2024-8-29

2. Javan K, Lialestani MRFH, Nejadhossein M. A comparison of ANN and HSPF models for runoff simulation in Gharehsoo River watershed, Iran. Model Earth Syst Environ. 2015;1: 41. doi:10.1007/s40808-015-0042-1

3. Vishwakarma DK, Kumar P, Yadav KK, Ali R, Markuna S, Chauhan S, et al. Evaluation of CatBoost Method for Predicting Weekly Pan Evaporation in Subtropical and Sub-Humid Regions. Pure Appl Geophys. 2024;181: 719–747. doi:10.1007/s00024-023-03426-4

4. Elbeltagi A, Al-Mukhtar M, Kushwaha NL, Al-Ansari N, Vishwakarma DK. Forecasting monthly pan evaporation using hybrid additive regression and data-driven models in a semi-arid environment. Appl Water Sci. 2022;13: 42. doi:10.1007/s13201-022-01846-6

5. Amiri E. Forecasting daily river flows using nonlinear time series models. J Hydrol (Amst). 2015;527: 1054–1072. doi: https://doi.org/10.1016/j.jhydrol.2015.05.048

6. Wang S, Peng H. Multiple spatio-temporal scale runoff forecasting and driving mechanism exploration by K-means optimized XGBoost and SHAP. J Hydrol (Amst). 2024;630: 130650. doi: https://doi.org/10.1016/j.jhydrol.2024.130650

7. Guo J, Liu Y, Zou Q, Ye L, Zhu S, Zhang H. Study on optimization and combination strategy of multiple daily runoff prediction models coupled with physical mechanism and LSTM. J Hydrol (Amst). 2023;624: 129969. doi: https://doi.org/10.1016/j.jhydrol.2023.129969

8. Xu Y, Lin K, Hu C, Wang S, Wu Q, Zhang J, et al. Interpretable machine learning on large samples for supporting runoff estimation in ungauged basins. J Hydrol (Amst). 2024;639: 131598. doi: https://doi.org/10.1016/j.jhydrol.2024.131598

9. Wang Y, Wang W, Xu D, Zhao Y, Zang H. A compound approach for ten-day runoff prediction by coupling wavelet denoising, attention mechanism, and LSTM based on GPU parallel acceleration technology. Earth Sci Inform. 2024;17: 1281–1299. doi:10.1007/s12145-023-01212-3

10. Guo H, Chen L, Fang Y, Zhang S. Model and application of annual river runoff prediction based on complementary set empirical mode decomposition combined with particle swarm optimization adaptive neuro-fuzzy system. Water Supply. 2023;23: 1760–1774. doi:10.2166/ws.2023.075

11. Kumar K, Singh V, Roshni T. Application of the PSO–neural network in rainfall–runoff modeling. Water Pract Technol. 2022;18: 16–26. doi:10.2166/wpt.2022.155

12. Katipoğlu OM, Sarıgöl M. Improving the accuracy of rainfall-runoff relationship estimation using signal processing techniques, bio-inspired swarm intelligence and artificial intelligence algorithms. Earth Sci Inform. 2023;16: 3125–3141. doi:10.1007/s12145-023-01081-w

13. Katipoğlu OM, Yeşilyurt SN, Dalkılıç HY, Akar F. Application of empirical mode decomposition, particle swarm optimization, and support vector machine methods to predict stream flows. Environ Monit Assess. 2023;195: 1108. doi:10.1007/s10661-023-11700-0

14. Hameed MM, Mohd Razali SF, Wan Mohtar WHM, Yaseen ZM. Examining optimized machine learning models for accurate multi-month drought forecasting: A representative case study in the USA. Environmental Science and Pollution Research. 2024. doi:10.1007/s11356-024-34500-6

15. Hameed MM, Masood A, Srivastava A, Abd Rahman N, Mohd Razali SF, Salem A, et al. Investigating a hybrid extreme learning machine coupled with Dingo Optimization Algorithm for modeling liquefaction triggering in sand-silt mixtures. Sci Rep. 2024;14: 10799. doi:10.1038/s41598-024-61059-6

15. Alomar MK, Khaleel F, Aljumaily MM, Masood A, Razali SFM, AlSaadi MA, et al. Data-driven models for atmospheric air temperature forecasting at a continental climate region. PLOS ONE. 2022;17: e0277079.

16. Döll P, Abbasi M, Messager ML, Trautmann T, Lehner B, Lamouroux N. Streamflow Intermittence in Europe: Estimating High-Resolution Monthly Time Series by Downscaling of Simulated Runoff and Random Forest Modeling. Water Resources Research. 2024;60: e2023WR036900. doi: https://doi.org/10.1029/2023WR036900

17. Zema DA, Parhizkar M, Plaza-Alvarez PA, Xu X, Lucas-Borja ME. Using random forest and multiple-regression models to predict changes in surface runoff and soil erosion after prescribed fire. Modeling Earth Systems and Environment. 2024;10: 1215–1228. doi:10.1007/s40808-023-01838-8

MATERIALS & METHOD

METHODOLOGY

• Consider adding brief explanations on why specific methods were chosen.

Reply: We provide a brief explanation of the main reasons for using these models in runoff forecasting. Please refer to page 13 (lines 246 to 258) for more details.

RESULTS

• Improve the logical flow and ensure that each finding is clearly explained.

Reply: We have improved the logical flow of the results section as suggested and have clearly presented the findings of this research. Please refer to pages 23, 26, 27, and 28, for details.

•Please give applied model parameters.

Reply: The model parameters are provided in the Appendix; please see Table S2.

CONCLUSIONS

• Clearly stating the most significant findings, with specific reference to the data.

• Offer more detailed and actionable recommendations based on the study's results.

• Highlight the study's contribution to existing research and its potential implications for theory and practice.

• Ensure that each paragraph is concise and directly contributes to the discussion.

Reply: We have revised the Conclusion section in accordance with your constructive comments:

- We have clearly explained the key findings, incorporating specific data as suggested.

- The major contributions of the study are stated as you recommended.

- We have highlighted the potential implications and practical applications of the research findings.

- Each paragraph of the Conclusion has been made concise and now directly contributes to the discussion, as suggested.

- Recommendations have been provided as advised.

Reviewer #2:

This paper proposes Extreme Gradient Boosting (XGBoost) and Deep Learning Models to predict the Monthly Runoff in a Glacierized Catchment. However, the quality of the paper should be improved a bit. Specific comments are as follows.

1. The graphical abstract/flow diagram is missing, it must be added.

Reply: A flow diagram has been added and explained. Please see Figure 5 on page 19.

2. The novelty statement and research gap should be highlights in innovative aspect of this study, specifically in abstract and Introduction.

Reply: The novelty of the current research has been highlighted in both the introduction and abstract sections, as suggested. Please see page 2 (Abstract) , and pages 6, and 7 (lines 133 – 146), in the introduction section.

3. The shortcomings of existing research should be summarized, and the innovation of this paper should be highlighted.

Reply: We have summarized the most essential shortcomings of existing research and highlighted the significance and innovation of our paper in the Introduction section. Please refer to pages 4 (lines, 75-85) and 6, and 7 (lines 133-146, 154-159).

4. As described in the paper, Extreme Gradient Boosting (XGBoost) and Deep Learning Models has been widely applied in the various fields, such as river stage forecasting (https://doi.org/10.1007/s13201-024-02103-8;

https://doi.org/10.1016/j.heliyon.2023.e16290), Drought Forecasting (https://doi.org/10.3390/w15040765) and long-term rainfall prediction (https://doi.org/10.1007/s00024-022-03189-4), Pan Evaporation (https://doi.org/10.1007/s00024-023-03426-4;
https://doi.org/10.1007/s13201-022-01846-6). However, several aspects in health monitoring, such as the reason for the adoption of Extreme Gradient Boosting (XGBoost) and Deep Learning Models in this paper is not clearly explained, it need to be appropriately supplemented.

Reply: We have improved our paper accordingly. Below is a list of some additional references included in the paper:

1. Al-Kubaisi MHD. Surface Runoff Estimation in Kubaisa Watershed Using SWAT, Western Desert, Iraq. Iraqi Geological Journal. 2024;57: 286–297. doi:10.46717/igj.57.2B.19ms-2024-8-29

2. Javan K, Lialestani MRFH, Nejadhossein M. A comparison of ANN and HSPF models for runoff simulation in Gharehsoo River watershed, Iran. Model Earth Syst Environ. 2015;1: 41. doi:10.1007/s40808-015-0042-1

3. Vishwakarma DK, Kumar P, Yadav KK, Ali R, Markuna S, Chauhan S, et al. Evaluation of CatBoost Method for Predicting Weekly Pan Evaporation in Subtropical and Sub-Humid Regions. Pure Appl Geophys. 2024;181: 719–747. doi:10.1007/s00024-023-03426-4

4. Elbeltagi A, Al-Mukhtar M, Kushwaha NL, Al-Ansari N, Vishwakarma DK. Forecasting monthly pan evaporation using hybrid additive regression and data-driven models in a semi-arid environment. Appl Water Sci. 2022;13: 42. doi:10.1007/s13201-022-01846-6

5. Amiri E. Forecasting daily river flows using nonlinear time series models. J Hydrol (Amst). 2015;527: 1054–1072. doi: https://doi.org/10.1016/j.jhydrol.2015.05.048

6. Wang S, Peng H. Multiple spatio-temporal scale runoff forecasting and driving mechanism exploration by K-means optimized XGBoost and SHAP. J Hydrol (Amst). 2024;630: 130650. doi: https://doi.org/10.1016/j.jhydrol.2024.130650

7. Guo J, Liu Y, Zou Q, Ye L, Zhu S, Zhang H. Study on optimization and combination strategy of multiple daily runoff prediction models coupled with physical mechanism and LSTM. J Hydrol (Amst). 2023;624: 129969. doi: https://doi.org/10.1016/j.jhydrol.2023.129969

8. Xu Y, Lin K, Hu C, Wang S, Wu Q, Zhang J, et al. Interpretable machine learning on large samples for supporting runoff estimation in ungauged basins. J Hydrol (Amst). 2024;639: 131598. doi: https://doi.org/10.1016/j.jhydrol.2024.131598

9. Wang Y, Wang W, Xu D, Zhao Y, Zang H. A compound approach for ten-day runoff prediction by coupling wavelet denoising, attention mechanism, and LSTM based on GPU parallel acceleration technology. Earth Sci Inform. 2024;17: 1281–1299. doi:10.1007/s12145-023-01212-3

10. Guo H, Chen L, Fang Y, Zhang S. Model and application of annual river runoff prediction based on complementary set empirical mode decomposition combined with particle swarm optimization adaptive neuro-fuzzy system. Water Supply. 2023;23: 1760–1774. doi:10.2166/ws.2023.075

11. Kumar K, Singh V, Roshni T. Application of the PSO–neural network in rainfall–runoff modeling. Water Pract Technol. 2022;18: 16–26. doi:10.2166/wpt.2022.155

12. Katipoğlu OM, Sarıgöl M. Improving the accuracy of rainfall-runoff relationship estimation using signal processing techniques, bio-inspired swarm intelligence and artificial intelligence algorithms. Earth Sci Inform. 2023;16: 3125–3141. doi:10.1007/s12145-023-01081-w

13. Katipoğlu OM, Yeşilyurt SN, Dalkılıç HY, Akar F. Application of empirical mode decomposition, particle swarm optimization, and support vector machine methods to predict stream flows. Environ Monit Assess. 2023;195: 1108. doi:10.1007/s10661-023-11700-0

14. Hameed MM, Mohd Razali SF, Wan Mohtar WHM, Yaseen ZM. Examining optimized machine learning models for accurate multi-month drought forecasting: A representative case study in the USA. Environmental Science and Pollution Research. 2024. doi:10.1007/s11356-024-34500-6

15. Hameed MM, Masood A, Srivastava A, Abd Rahman N, Mohd Razali SF, Salem A, et al. Investigating a hybrid extreme learning machine coupled with Dingo Optimization Algorithm for modeling liquefaction triggering in sand-silt mixtures. Sci Rep. 2024;14: 10799. doi:10.1038/s41598-024-61059-6

15. Alomar MK, Khaleel F, Aljumaily MM, Masood A, Razali SFM, AlSaadi MA, et al. Data-driven models for atmospheric air temperature forecasting at a continental climate region. PLOS ONE. 2022;17: e0277079.

16. Döll P, Abbasi M, Messager ML, Trautmann T, Lehner B, Lamouroux N. Streamflow Intermittence in Europe: Estimating High-Resolution Monthly Time Series by Downscaling of Simulated Runoff and Random Forest Modeling. Water Resources Research. 2024;60: e2023WR036900. doi: https://doi.org/10.1029/2023WR036900

17. Zema DA, Parhizkar M, Plaza-Alvarez PA, Xu X, Lucas-Borja ME. Using random forest and multiple-regression models to predict changes in surface runoff and soil erosion after prescribed fire. Modeling Earth Systems and Environment. 2024;10: 1215–1228. doi:10.1007/s40808-023-01838-8

---

## [Decision Letter · Decision Letter 1]

28 Feb 2025

Forecasting Monthly Runoff in a Glacierized Catchment: A Comparison of Extreme Gradient Boosting (XGBoost) and Deep Learning Models.

PONE-D-24-40156R1

Dear Dr. Salem,

We’re pleased to inform you that your manuscript has been judged scientifically suitable for publication and will be formally accepted for publication once it meets all outstanding technical requirements.

Kind regards,

Shicheng Li

Academic Editor

PLOS ONE

Additional Editor Comments (optional):

Reviewers' comments:

Reviewer's Responses to Questions

**Comments to the Author**

1. If the authors have adequately addressed your comments raised in a previous round of review and you feel that this manuscript is now acceptable for publication, you may indicate that here to bypass the “Comments to the Author” section, enter your conflict of interest statement in the “Confidential to Editor” section, and submit your "Accept" recommendation.

Reviewer #3: All comments have been addressed

Reviewer #4: All comments have been addressed

Reviewer #5: All comments have been addressed

2. Is the manuscript technically sound, and do the data support the conclusions?

Reviewer #3: (No Response)

Reviewer #4: Yes

Reviewer #5: Yes

3. Has the statistical analysis been performed appropriately and rigorously? 

Reviewer #3: (No Response)

Reviewer #4: Yes

Reviewer #5: Yes

4. Have the authors made all data underlying the findings in their manuscript fully available?

Reviewer #3: (No Response)

Reviewer #4: Yes

Reviewer #5: Yes

5. Is the manuscript presented in an intelligible fashion and written in standard English?

Reviewer #3: (No Response)

Reviewer #4: Yes

Reviewer #5: Yes

6. Review Comments to the Author

Reviewer #3: (No Response)

Reviewer #4: As is been already reviewed and all the queries answer by authors successfully. Now it can be published.

Reviewer #5: The paper titled “Forecasting Monthly Runoff in a Glacierized Catchment: A Comparison of Extreme Gradient Boosting (XGBoost) and Deep Learning Models” explores monthly runoff forecasting in a glacierized catchment. The authors utilize novel forecasting models along with a robust statistical assessment method, Average Absolute Error of Turning Points (AETP). This study could be a valuable addition to the existing literature. However, there are several issues that should be addressed:

1. The authors should better emphasize the significance of monthly runoff forecasting for drought and flood monitoring in the Introduction.

2. In the Results and Discussion, the authors analyze model accuracy only at the monthly and seasonal levels. A more detailed analysis for each month is needed to highlight which months AI models forecast runoff more accurately.

3. It is advisable to apply combined accuracy metrics rather than relying solely on single statistical measures such as RMSE and MAE to assess model performance more effectively.

4. The study should include limitations and future recommendations to provide a more comprehensive discussion.

5. The article could benefit from considering additional recent relevant references in the state-of-the-art analysis on runoff forecasting, such as:

https://doi.org/10.1016/j.rineng.2024.102104

https://doi.org/10.1007/s00477-024-02760-w

https://doi.org/10.1016/j.jhydrol.2024.132175

7. PLOS authors have the option to publish the peer review history of their article (what does this mean? ). If published, this will include your full peer review and any attached files.

**Do you want your identity to be public for this peer review?** For information about this choice, including consent withdrawal, please see our Privacy Policy .

Reviewer #3: No

Reviewer #4: **Yes: ** Dr Bibhuti Bhusan Sahoo

Reviewer #5: No

---

## [Editor Report · Acceptance letter]

PONE-D-24-40156R1

PLOS ONE

Dear Dr. Salem,

I'm pleased to inform you that your manuscript has been deemed suitable for publication in PLOS ONE. Congratulations! Your manuscript is now being handed over to our production team.

Kind regards,

on behalf of

Dr. Shicheng Li

Academic Editor

PLOS ONE